# Optogenetic mapping of feeding and self-stimulation within the lateral hypothalamus of the rat

**Kevin R. Urstadt**[ID]¤*, **Kent C. Berridge**[ID]

Psychology Dept., University of Michigan, Ann Arbor, Michigan, United States of America

¤ Current address: Cognitive Science Dept., Occidental College, Los Angeles, California, United States of America
* kurst001@gmail.com

**Data Availability Statement:** The raw data used in this project are stored on the Open Science Framework repository (DOI 10.17605/OSF.IO/9TUD7).

## Abstract

The lateral hypothalamus (LH) includes several anatomical subregions involved in eating and reward motivation. This study explored localization of function across different LH subregions in controlling food intake stimulated by optogenetic channelrhodopsin excitation, and in supporting laser self-stimulation. We particularly compared the tuberal LH subregion, the posterior LH subregion, and the lateral preoptic area. Local diameters of tissue optogenetically stimulated within the LH were assessed by measuring laser-induced Fos plumes and Jun plumes via immunofluorescence surrounding optic fiber tips. Those plume diameters were used to map localization of function for behavioral effects elicited by LH optogenetic stimulation. Optogenetic stimulation of the tuberal subsection of the LH produced the most robust eating behavior and food intake initially, but produced only mild laser self-stimulation in the same rats. However, after repeated exposures to optogenetic stimulation, tuberal LH behavioral profiles shifted toward more self-stimulation and less food intake. By contrast, stimulation of the lateral preoptic area produced relatively little food intake or self-stimulation, either initially or after extended stimulation experience. Stimulation in the posterior LH subregion supported moderate self-stimulation, but not food intake, and at higher laser intensity shifted valence to evoke escape behaviors. We conclude that the tuberal LH subregion may best mediate stimulation-bound increases in food intake stimulated by optogenetic excitation. However, incentive motivational effects of tuberal LH stimulation may shift toward self-stimulation behavior after repeated stimulation. By contrast, the lateral preoptic area and posterior LH do not as readily elicit either eating behavior or laser self-stimulation, and may be more prone to higher-intensity aversive effects.

## Introduction

The lateral hypothalamus (LH) has been considered a powerful regulator of food ingestion and reward-seeking motivation for over 60 years [1]. Today the LH remains a prime target of research into obesity, anorexia, and reward-related motivational dysfunctions [2]. Early

**Funding:** The author KCB received grants from the National Institutes of Health (DA015188 and MH63649). The author KRU received a training grant from the National Institutes of Health (DC00011). Additional information about such grants can be found through https://grants.nih.gov/grants/oer.htm. The funders had no role in study design, data collection and analysis, decision to publish, or preparation of the manuscript.

**Competing interests:** The authors have declared that no competing interests exist.

decades of research used lesion, electrical stimulation, electrophysiological recording, and intracranial microinjection techniques, whereas many contemporary studies have shifted to optogenetic, DREADD, and optical imaging techniques.

The LH was first anatomically described nearly a century ago [3,4]. Electrolytic LH lesions were soon known to result in aphagia, adipsia, and sensory neglect behaviors [5–8]. Further, in early studies large LH lesions also produced pathological excessive 'disgust', evident as gapes, headshakes and chin rubs that are normally elicited only by bitter or other unpalatable tastes, becoming elicited by the taste of sucrose [7,9,10]. However, subsequent excitotoxin lesion studies showed that the site where neuron loss produced 'disgust' is actually in caudolateral ventral pallidum, anterior to LH, and not in LH itself [11,12]. LH lesions that do not damage the ventral pallidum, as well as neurotoxic destruction of dopamine fibers of passage through LH, do produce aphagia and sensory neglect, but not excessive 'disgust' [7,8,11,13–16]. These lesion data paint a strong necessity role for the LH in intake, while neighboring regions and fibers passing through the LH control a broader range of reward, disgust, and motivation-related functions.

Other electrical stimulation experiments demonstrated that LH activation elicits eating, drinking and other natural motivated behaviors [17]. Additionally, rats were typically willing to self-stimulate or work to activate the same LH electrodes, implicating the LH in both hunger and reward [17–21]. However, for most LH stimulation effects, there was not clear localization of function within the LH [17]. Conceivably, stimulation of fibers of passage by an LH electrode may also play a role in behavioral effects, which could possibly obscure localization of function within subregional clusters of LH neurons [22,23], making it difficult to specify the relative contribution of intrinsic LH neurons. The possibility of localization of function within LH for neuronal stimulation effects could yet emerge, if explored with modern techniques, such as optogenetic stimulation of neurons in particular LH subregions.

Beyond the question of subregional LH differences in localization of function, another important issue is the permanence versus malleability of behavioral effects of LH stimulation. For example, using electrode stimulation in the LH, Valenstein and colleagues [24] showed that the behavior evoked from rats by LH stimulation could change over time, due to repeated experiences with electrical LH stimulation. For example, some LH stimulation sites did not initially evoke eating, but subsequently did after rats received prolonged exposures to electrode stimulation overnight while in a food deprived state [25]. For other rats, the dominant type of behavior elicited by LH stimulation switched from LH-evoked eating to LH-evoked drinking in the same rats, after repeated experiences with LH stimulation in which food targets were removed but water was available [26]. Later when food was returned and a choice was available, those rats remained stimulation-bound drinkers. Such reports suggest the possibility that repeated experience might also possibly change the type of behavior evoked by LH optogenetic neuronal stimulation.

To address these issues, we utilized optogenetic channelrhodopsin (ChR2) stimulation to excite neurons in specific LH subregions of different rats. Excitation of LH neurons in specific subregions, without stimulating fibers of passage, was assessed both for laser self-stimulation and for stimulation-bound increases in eating. We also mapped potential localization of function. In order to identify how large a subregion of LH was activated by ChR2 stimulation, which is information required to assign localization of function, we measured immunofluorescence of local Fos plumes induced by laser illumination around an optic fiber tip, and compared their sizes to those of corresponding Jun plumes; both are known as immediate early gene products (IEG). Additionally, we assessed if any laser-evoked behaviors changed after a phase of extended experience with LH optogenetic stimulation.

## Methods and materials

### Subjects

Pair-housed adult Sprague-Dawley rats (*Rattus norvegicus*; N = 32; male $N_{ChR2}$ = 10; male $N_{eYFP}$ = 6; female $N_{ChR2}$ = 11; female $N_{eYFP}$ = 5) weighing 250–400 g at the time of surgery were maintained on a reverse 12 hr light-dark cycle at 21˚C. Rats were always provided ad libitum access to standard Purina rat chow and to water in home cages unless otherwise specified. Males and females were housed in separate rooms, tested separately, and test chambers were cleaned between uses to prevent sexual odor cues from influencing behavior. Animal use procedures were approved by the Institutional Animal Care and Use Committee at the University of Michigan. A brief depiction of the experimental timeline can be seen in Fig 1.

### Surgery

Rats were anesthetized with ketamine (100 mg/kg, i.p.) and xylazine (7 mg/kg, i.p.) and were given atropine (0.04 mg/kg, i.p.), carprofen (5 mg/kg, s.c.), cefazolin (60 mg/kg, s.c.), and isotonic saline (2 mL, s.c.) during surgery to prevent respiratory occlusion, inflammation, infection, and dehydration, respectively. Using a stereotaxic apparatus, rats received surgical microinjections of either an AAV5 virus containing genes for channelrhodopsin (ChR2) for optogenetic stimulation, an enhanced yellow fluorescence protein (eYFP) reporter, and hSyn promoter (AAV5 hSyn-ChR2-eYFP), or of an optically-inactive control virus, containing genes only for hSyn promoter and eYFP reporter (AAV5 hSyn-eYFP). Viral vectors were obtained by permission of Karl Deisseroth from the University of North Carolina Viral Vector Core. Microinjections were aimed into the LPO or LH at various sites at staggered coordinates, ranging 1.0 to 4.0 mm posterior, 1.0 to 2.5 mm lateral, and 8.0 to 9.5 mm ventral from bregma. Virus solution was infused via 28-gauge microinjector tips connected to PE 20 tubing attached to 5 μL Hamilton syringes in an injector pump. A 0.75 μL virus volume per hemisphere was

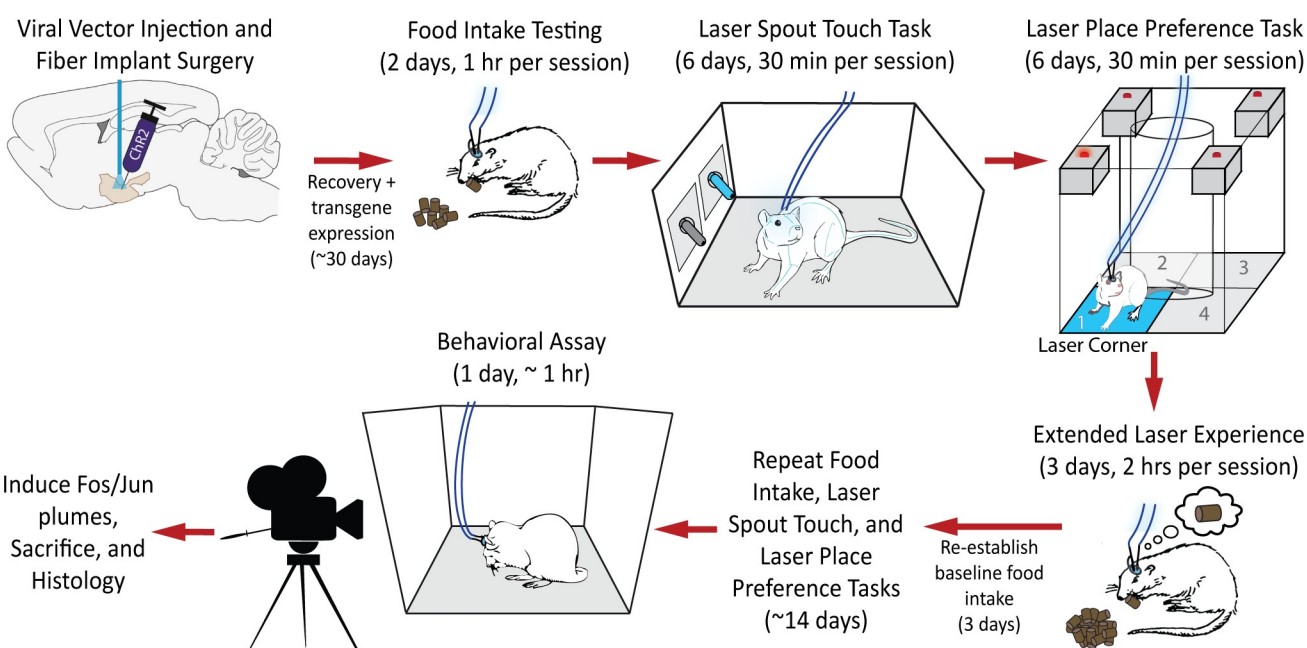

**Fig 1. Experimental timeline and design.** All subjects underwent the following process depicted here. Each new task is started the following day (denoted by a red arrow) unless a longer time elapses as stated.

bilaterally infused over 10 min, followed by an additional 10 min diffusion wait period before microinjection cannulas were removed. Bilateral fiber optic implants (core diameter 300 μm) were implanted into the same LH sites as the virus microinjections, except placed 0.4 mm more dorsal to the microinjection so as to illuminate the virus-infected zone beneath. Jeweler's screws were placed in the skull as anchors, and dental cement was applied to secure the implants. Atipamezole (1 mg /kg, s.c.) was administered after surgery to hasten post-operative awakening. Rats were allowed to recover and incubate virus for one month to allow sufficient ChR2 expression before behavioral testing began.

### Laser delivery

Laser light at 473 nm wavelength was always applied at 25 Hz (15 ms on, 25 ms off per pulse) for pulse train lengths specified in each experiment. Laser intensity was set to 2–3 mW/mm$^2$ emanating from the fiber tip for all experiments, except when intensity was manipulated for comparison of multiple intensities as described below. For all behavioral tests, laser illumination was delivered via optic fiber through a bifurcating rotary joint (Doric) attached to a weighted lever arm positioned over each testing box. Fiber optic cables (core diameter 300 um), assembled in-house, were attached to the rotary joint via FC connectors, and attached to rat fiber implants via tight-fitting brass sleeves. Stretching thread was tied to the middle and top of each cable to keep excess cable length out of reach of the rats. Sleeves were obscured with heat-shrink tubing to minimize external laser light escape and visibility to the rats during tests.

### Food intake tests

Each rat was tested for food intake for 1 h in white ambient lighting in sessions in clear cage bottoms with pre-weighed Purina rat chow pellets, a Richter tube containing water, a 2-cm wide wooden block for potential gnawing, and a 1-inch layer of corncob bedding. Laser pulse trains were applied in cycles of 5 s on, 15 s off. Laser sessions and non-laser sessions were performed on consecutive days in a counterbalanced order across rats. Video-recorded eating and drinking behaviors were subsequently scored in slow-motion off-line analysis.

### Active laser self-stimulation via touching metal spouts

For 3 consecutive days in 30 min sessions, each rat was tested in Med Associates operant chambers with sound reduction and red ambient lighting. Two metal spouts (similar to water sipper spouts in home cage) protruded though the back wall, positioned about 5 cm apart. A circuit connected the metal floor grid to the both spouts such that rats touching either spout completed the circuit, and the spout contacts were recorded. To earn laser self-stimulation, each contact with one designated empty metal spout triggered a brief train of laser illumination (1 s pulse train at 25 Hz, 2 mW for each contact). Contact with the other spout did nothing and served as a control measure of touches due to general activity, exploration, etc. The assignment as "laser-delivering spout" versus inactive-control spout remained consistent across three consecutive days for each rat but was counterbalanced between rats.

If a preference had been shown in the series above (defined as minimum of 100 laser spout touches, with non-laser touches being no more than half the laser touches), the rat was retested in a second series of tests for three additional consecutive days but with the laser assignment to the spouts reversed, in order to confirm motivated self-stimulation by tracking the change in active spout location.

## Laser place preference/avoidance tests

In a place-based laser self-administration test, rats could earn laser stimulations simply by entering a location or remaining there [27]. This was modeled after the original place-based discovery of brain self-stimulation electrodes by Olds and Milner [19]. On 3 consecutive days, each rat was tested for 30 min sessions in red ambient light in a large plexiglass 4-cornered chamber, in which each corner had a motion-detector to detect entries or continual movements while the rat remained in that corner, as described previously [27]. One corner was designated to deliver laser as 1 s trains of 25 Hz, 2 mW laser stimulation upon entry or subsequent movement. Each rat was assigned a laser corner that remained consistent across 3 testing days. After this three-day testing period, responses were assessed. Subsequently, all rats were re-tested in the same chamber for three additional days but with the corner diagonally opposite to the original now assigned laser. Corner entries were recorded by a Matlab program. The number of motion-detector triggers for each corner are represented as a percentage of the total across all corners. This normalization was done to match subjects with different overall activity levels. Rats were deemed to prefer their laser corner if >30% of 4-corner entries occurred in the laser corner, whereas other rats were deemed to avoid their laser corner when <20% of corner entries occurred in the laser corner.

We utilized this place preference/avoidance test in addition to the spout self-stimulation test as each evaluates the motivational valence of laser stimulation in somewhat different ways. The place preference/avoidance test tends to deliver more prolonged or repeated laser stimulation, because a new 1 s stimulation is triggered by each movement within the laser-administering corner. This provides a rat with an easy way to bask in prolonged laser stimulation if it is rewarding. The place test also allows the detection of avoidance of aversive laser properties, again especially if these emerge with more dense exposure to laser stimulation. Finally, the place test is sensitive to Pavlovian conditioning of preference or avoidance to the location as conditioned stimulus. The spout self-stimulation test, on the other hand, gives a rat more precise instrumental control of laser stimulation. It requires a new active instrumental response (spout touch) to trigger a laser stimulation. It also provides the rat with greater control over the amount and timing of stimulation. The spout test typically delivers less dense laser stimulation, allowing best detection of reward/incentive effects, especially if laser becomes more negatively aversive with greater density. And the spout is directly sensitive to detecting traditional instrumental response reinforcement, that is, strengthening the probability of an immediately preceding and action. For these reasons, we believe that both tests of self-stimulation have value. And when both give the same answer, they establish the generality of positive incentive effects of laser by showing that self-stimulation is not limited to the parameters of one particular paradigm.

## Food intake and spontaneous behavior tests

Each rat was video recorded for food intake in a short-term free access test, conducted during one 20 min session per day in white lighting. Chow pellets, water-filled Richter tubes, wooden gnawing blocks, and bedding were provided. Laser stimulation was cycled for 20 s on, 90s off multiple times. Behaviors occurring in the presence of laser versus baseline levels without laser were subsequently compared to assess laser-evoked behaviors. Laser intensity was either: low (0.2), medium (1), high (5), and very high (10 mW/mm$^2$) power stimulation. A given intensity was chosen randomly to start the day, and was repeated for 5 min. Then intensity changed randomly to one of the remaining levels and repeated for 5 min, changed again and repeated for 5 min, and changed again and repeated for 5 min. Thus, each of the four intensities was tested once per day in its own 5-min session. Our plan was to end sessions immediately if any

behavioral signs of seizure activity were observed, such as repetitive mouth or limb twitching or signs of distress such as vocalizations or escape attempts. However, no seizure activity was observed either during these behavioral assays nor during any of the other tasks used in this study.

## Scoring of video-recorded behaviors

Offline behavioral analysis was conducted at a playback of ½ - ¼ real-time speed. Behaviors were labeled as the following: chow eating, chow carrying, block carrying, block gnawing, drinking, hyperkinetic treading (exaggerated kinematics and repeated persistence of this form), defensive treading (forepaws oriented palm forward, without body weight, so as to push bedding forward), mouth gaping, chin rubbing, head shaking, cage crossing, rearing, grooming, turning, digging, running, jumping, stationary and sleeping.

Gaping, chin rubbing, head shakes, turning, and jumping were counted as discrete events, and every occurrence was counted. All other behaviors were scored as duration of time spent engaged in that behavior and converted to normalized scores. Normalization was done by dividing by the cumulative duration of the behavior by the cumulative length of the scoring period category (laser, following, or interim) at that laser intensity, and then multiplying by 20 to scale the resulting values up to match magnitudes of the individually counted behaviors (gaping, jumping, etc.). Separate scores were calculated for each laser period (20 s), period immediately following laser (20 s), and interim pre-laser (70 s) period.

## Extended laser experience with food while hungry phase

After previously being tested for both food intake and laser self-stimulation elicited by optogenetic LH stimulation, rats were given extended optogenetic experience, and then retested again. Following a laser-experience procedure similar to that used by Valenstein and Cox [25], for 3 consecutive days, rats that were food-deprived for 22 hours (water always ad lib) were exposed for 2 hours to intermittent laser illuminations in the presence of food, water, and wooden blocks (25 Hz, 2 mW laser for a 30 s period once every 5 min; during the rat's dark period while in constant red light). Food intake was monitored during these sessions, chow was weighed before and after, and body weight was recorded before and after the session. Ad lib access to food was returned immediately after the third session, and a 24-hr ad lib food access rest day prior to next test was allowed to re-equilibrate eating and body weight. Following this rest day, rats were retested in the food intake, spontaneous behavior, and laser self-stimulation tests as described above. The question was whether prolonged exposures to intermittent laser had altered the profile of behavioral responses elicited by LH ChR2 stimulation.

## Identification of LH sites and diameters of stimulated tissue; Local Fos/Jun plumes

Local plumes of elevated Fos immunohistochemical expression immediately surrounding an illuminated optic fiber provide a measure of the diameter of local neural tissue directly excited by ChR2 stimulation [27,28]. Here we added analyses of Jun expression to assess local Jun plumes of elevation for comparison to Fos plumes in order to confirm whether similar tissue diameter measures are given by expression of different Fos vs Jun immediate early genes.

Two hours prior to perfusion, each rat was given 25 Hz, 3 mW/mm$^2$ laser stimulation cycled 5 s on, 15 s off for 30 min in a food-intake testing chamber. However, no food, water, or gnawing blocks were provided during this laser stimulation; only bedding was present. Ninety minutes after the end of this session, the rat was anesthetized with a fatal overdose of sodium pentobarbital and transcardially perfused with isotonic saline with 18.3 mM procaine HCl and

0.01 M sodium phosphate buffer (PB) at pH 7.4 followed by buffered 4% formaldehyde. Brains were extracted, post-fixed in formaldehyde solution for 2 days, incubated in 25% sucrose in 0.1 M PB for 4 days, and cryostat sectioned into 50 um sections in 3 series. These sections were collected in an antifreeze solution (50% v/v 0.1M PB, 30% v/v ethylene glycol, 19% w/v sucrose, 1% polyvinylpyrrolidone w/v) and stored at -20˚C until used.

One of every 3 tissue sections underwent immunohistochemistry for both Fos (red fluorescence) and Jun proteins (blue fluorescence; pseudocolored as green for visibility in figures below) to allow comparison of plume diameters and intensities, and assessment of overlap in identical neurons (green fluorescence expressed by eYFP introduced from virus transfection) in the same section. The other two sections of every three were retained as backups. The following rinses were done with 0.1 M PB for 15 min each. Antibodies and chemicals were diluted in 2.5% normal donkey serum, 0.2% Triton X-100, and 0.05% sodium azide in 0.1 M PB unless stated otherwise. Sections were rinsed in buffer with azide, incubated in 1% H2O2 for 15 minutes to neutralize endogenous peroxidases, rinsed, incubated in goat anti-c-fos (0.2 ug/mL, sc-52, Santa Cruz Biotech) and rabbit anti-c-jun (0.5 ug/mL, sc-1694, Santa Cruz Biotech) overnight at room temperature, rinsed, incubated in biotinylated donkey anti-goat (1:300, #705-005-147, Jackson ImmunoResearch) for 2 hrs, rinsed, incubated in streptavidin-conjugated horseradish peroxidase (1:300 in rinse solution, #016-030-084, Jackson ImmunoResearch) for 1.5 hrs, rinsed, incubated in rhodamine-tyramide (1 ug/mL in distilled water with 0.003% H2O2) for 20 min, rinsed in buffer with azide for 15 min, incubated in 1% H2O2 for 15 min (to deactivate first round of HRP), rinsed, incubated in horseradish peroxidase donkey anti-rabbit (1:300 in rinse solution, #A16035, ThermoFisher) for 2 hours, rinsed, incubated in aminomethylcoumarin-tyramide (AMCA-tyramide, 3.33 ug/mL in distilled water with 0.003% H2O2) for 20 min, rinsed, mounted on glass slides, dried, and coverslipped with a polyvinyl alcohol mounting medium made in-house.

## Histological evaluation of Fos, Jun, and virus expression

Sections were imaged using a Leica epifluorescent microscope with a digital camera. AMCA (c-jun), eYFP (ChR2), and rhodamine (c-fos) were visualized using Leica bandpass filter cubes A4, L5, and TX2, respectively. Surveyor software (Leica) was used to tile and stitch images into mosaics, and these mosaics were used for IEG counting. Fiber tip tracts and subsequently centers of Fos plumes were localized to specific coordinates on atlas pages [29]. This section-to-page matching was done by scaling pages in G.I.M.P., setting them to 50% opacity, and overlaying them onto section photomicrographs such that tissue landmarks (consisting of the fornix, optic tract, mamillothalamic tract, third ventricle, and cerebral penduncles) matched up to the atlas page in approximate size and inter-landmark distance. Laser-induced Fos and Jun plumes around the tips of fiber optic implants were measured similarly to previous work [28,30]. Fos plumes were assessed by overlaying an 8-direction radial arm grid, with 50 μm x 50 μm boxes emanating from underneath the tip tract's centerpoint in each direction, on to photomicrographs of brain tissue centered at the fiber tip. Intensity of IEG expression at points on the grid was counted as the number of stained neuron nuclei per 50 $\mu m^2$ box at that designated point (see [30] for example). Fos or Jun counts per each grid box were averaged within groups. Elevation in IEG expression was determined by comparing ChR2 laser group IEG counts versus control IEG counts measured at the same grid point in illuminated eYFP laser group. The average plume radius for the ChR2 laser group was constructed to reveal the size of directly stimulated tissue, and plume-sized symbols for each rat were placed on the corresponding location in atlas pages to create maps of ChR2-elicited behavioral effects in figures below.

As an additional metric for distinguishing laser-induced neuronal activation, average staining intensity of Fos-positive nuclei was assessed using ImageJ (National Institutes of Health) for three rats from each of the following groups: ChR2 with laser, ChR2 without laser, and eYFP with laser. The intensity difference was only apparent in the Fos plumes, because the Fos stain was even more amplified compared to the Jun stain, allowing only Fos to reveal this bright nuclei vs dim nuclei difference. Thus, intensity differences between different Jun-expressing neurons were not detected here. Areas within the Fos plume were selected for comparison to control baseline areas randomly selected far away from the optic fiber. Images of these areas were cropped, converted to 256 shade 8-bit grayscale mode, equally thresholded to highlight Fos-positive nuclei above the background and the thresholded images were converted to outlines. The "mean gray value" option was selected in the set measurements dialog, and the aforementioned outlines were used to direct ImageJ's analyze particles function on the original grayscale image. Average intensity was determined for six random Fos-positive nuclei inside versus outside the plume in the same brain section.

## Data structuring and statistical analysis

Virus and fiber placements in the LH or LPO were categorized into anatomical subregions along stereotaxic dimensions—anteroposterior (AP), mediolateral (ML), or dorsoventral (DV). Five AP subdivisions were distinguished between 1.0 and 4.0 mm posterior from bregma, specifically anterior LPO (aLPO; 1.0 to 1.6, posterior LPO (pLPO: 1.61 to 2.2), anterior LH (aLH; 2.21 to 2.8), tuberal LH (tLH: 2.81 to 3.4), and posterior LH (pLH; 3.41 to 4.0). Four ML subdivisions were distinguished bilaterally between 1.0 and 2.6 mm lateral from bregma (1.0 to 1.4, 1.41 to 1.8, 1.81 to 2.2, 2.21 to 2.6). Three DV subdivisions were distinguished between 8.0 and 9.2 mm ventral from bregma (8.0 to 8.4, 8.41 to 8.8, 8.81 to 9.2). Statistics were performed using SPSS 16.0 software. Behavioral effects associated with each site were compared across laser and non-laser conditions for the same rat as a within-subject factor. Behavioral changes evoked over time in the same rats from initial tests versus after extended laser exposures were also compared as a within-subject factor. Anatomical placement and sex were compared as between-subjects factors.

When significant effects ($p < 0.05$) were determined by ANOVAs, paired t-tests were employed to follow up for specific comparisons. Two-tailed tests were used for all comparisons except for laser versus non-laser comparisons in food intake and LSS experiments, which used one-tailed tests because the baselines in these experiments were low or near zero. Along with t-test results, means and standard error of the means are provided below. Effect sizes were determined using eta squared ($\eta^2$; sum of squares for the effect divided by total sum of squares) for ANOVAs, Cohen's d for t-tests with equal variance, and Hedges' g for t-tests with unequal variance. For the place-based self-stimulation results, significant anatomical location x corner interactions were followed up with one-way ANOVAs within each corner and each experience condition, followed by post-hoc Tukey's HSD pairwise testing. Unequal variances between ChR2 and eYFP subject groups were assessed by significant results of Levene's test for equality. These groups were compared via independent samples t-tests when they had equal variances and via Welch's t-test when they had unequal variances (indicated by the mention below of Levene's p; alpha < 0.05).

Significant effects and changes in strong vs weak/non-responses in food intake and self-stimulation tasks were determined using non-parametric statistics. To determine significant grouping of strong vs weak responses, binomial tests were used with the expected proportion of strong responders set to 0.1. Pre versus post extended laser experience differences were evaluated with Wilcoxon signed ranks testing. Anteroposterior subgroups were compared using

Kruskal-Wallis tests with the effect size measurement epsilon squared ($\varepsilon^2 = \chi^2 / ((N^2-1) / (N + 1)$) [31], followed by post-hoc Mann-Whitney U pairwise tests with $r^2$ (where $r^2 = Z^2 / N$) as a measure of effect size. Binomial and Wilcoxon tests utilized exact p values whereas Kruskal-Wallis and Mann-Whitney tests used asymptotic p values. All p values are two-tailed unless stated otherwise.

To determine laser versus non-laser effects in behavioral assays, a one-way within-subjects repeated measures ANOVA was performed for each laser power output for laser, following laser, and interim scoring periods as defined above, and significant results were followed with Fisher's pairwise testing. In order to determine differences between groups, laser difference scores were first calculated by subtracting the interim period score from the laser period score for a specific behavior at a specific laser power; these values reflected relative changes from baseline for each rat. Then, to detect significant differences between groups within AP, ML, or DV strata, a one-way ANOVA was run on these laser difference scores followed by pairwise testing of significant effects with Fisher's pairwise testing. Sex differences were assessed directly with independent samples t testing of laser difference scores. Interaction effects of location or sex and scoring period were assessed using a two-way mixed ANOVA, and significant effects were followed with independent samples t testing. Independent samples t tests determined significant differences between eYFP and ChR2 groups (without subdivision by sex or location). Two-way between-subjects ANOVAs were conducted to determine significant interactions effects of virus type and location or sex on laser difference scores, then post-hoc independent samples t tests were conducted to determine which subgroups significantly differed.

For independent samples t tests assessing differences between sexes or locations, two-tailed p values were used. In several other cases, one-tailed p values were used. These included any behavior that was near zero at baseline (eating, drinking, gnawing, chow/toy carry, digging, running, gaping, treading, headshaking, chin rubbing, jumping), and within these behaviors, assessing the difference of laser versus interim scoring periods and ChR2 subject scores versus eYFP subject scores.

Two-tailed Pearson's r correlational tests were run to assess potential correlation between individual performance on self-stimulation and food intake tests. These correlations were run both before and after extended laser experience. Finally, correlations were tested between laser power and behavioral scores to assess dose-response relationships. Alpha was set to 0.05 for these correlations.

For immunohistochemical results, repeated measures ANOVA was used to determine significant differences in Fos staining intensity inside versus outside the local IEG plume, as well as to compare IEG expression in ChR2 with laser, ChR2 without laser, and eYFP with laser groups. Post-hoc pairwise testing was conducted when significant differences were detected, and all tests used a 0.05 alpha criterion. Correlations were also assessed between Fos and Jun counts for grid box points on radial arms in laser conditions, to assess agreement between the two markers of neuronal activation.

## Results

### Fiber placement and IEG plume analysis

Laser stimulation of ChR2 virus in LH produced outer local plumes of >200% increases in both Fos and Jun expression within a ~0.3 mm radius of the fiber tip, and even stronger >250% inner plumes within a ~0.15 mm radius of the tip (Figs 2–4). Intensities of Fos and Jun expression in each radial arm box were highly correlated together (r(80) = 0.771, p < 0.001), suggesting that both Fos and Jun plumes reflected similar underlying neuronal activation. This similarity helps give confidence that Fos or Jun plume intensities and diameters do accurately

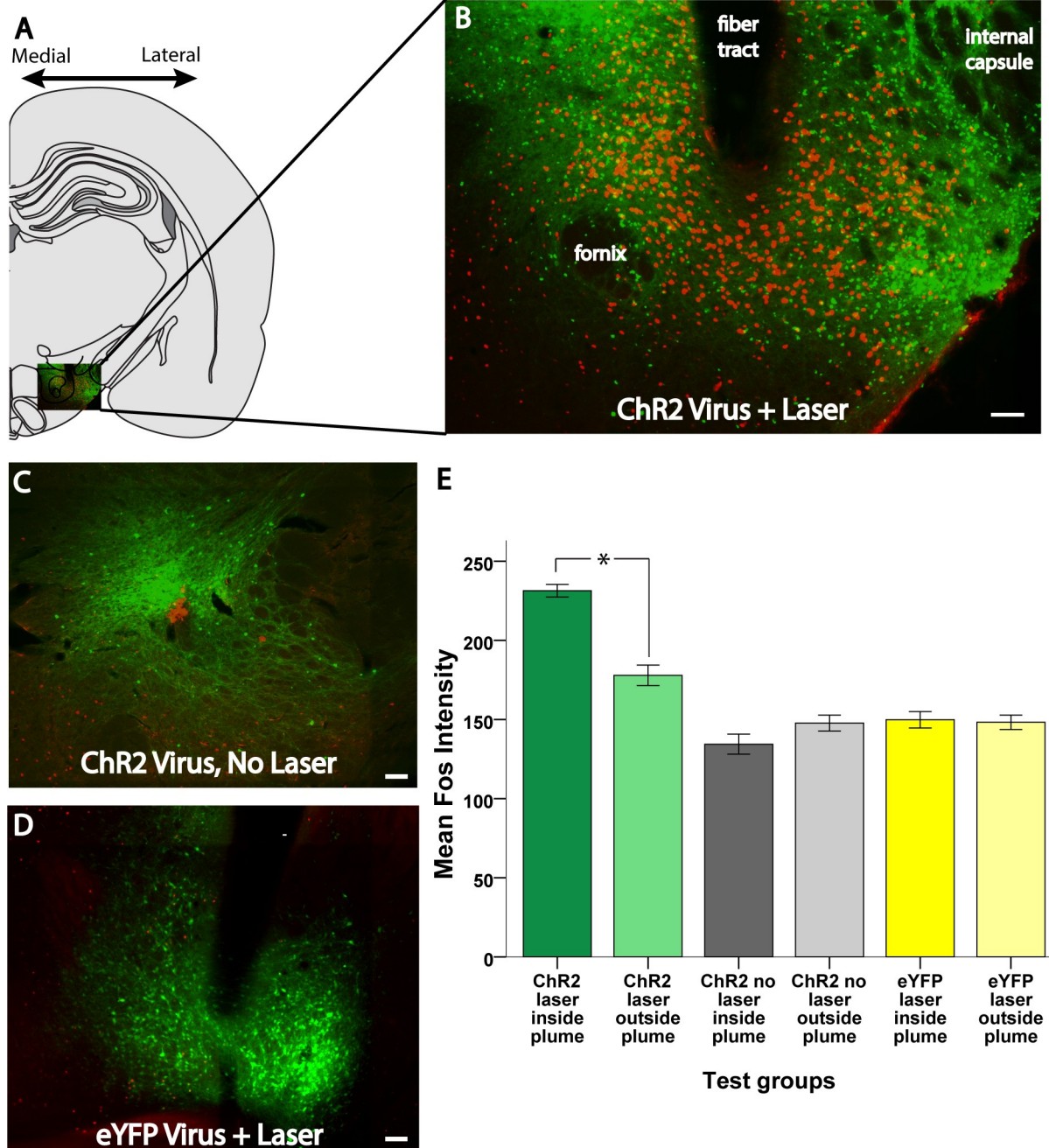

**Fig 2. Examples of virus expression (green) and Fos-like immunoreactivity (red) in different treatment conditions.** A: Hemisection of a coronal map modified from a rat brain atlas (29). B: an inset of the lateral hypothalamus expanded out, giving an example of a robust ChR2 laser-induced Fos plume. C, D: Examples of virus and Fos expression in animals that received ChR2 virus without laser stimulation and eYFP virus with laser stimulation. E. Average intensity of Fos granules inside vs outside plumes for each experimental group and condition. Scale bars represent 100 μm unless otherwise stated. *: p < 0.05.

reflect the spread of optogenetic neuronal excitation induced in tissue surrounding an illuminated optic fiber tip. Notably, the radii of Fos and Jun plumes were typically not much more than one-half the extent of virus infection (0.3 mm average radius Fos/Jun plumes versus 0.5 mm average radius virus infection; Fig 2B), suggesting that laser illumination neurobiologically

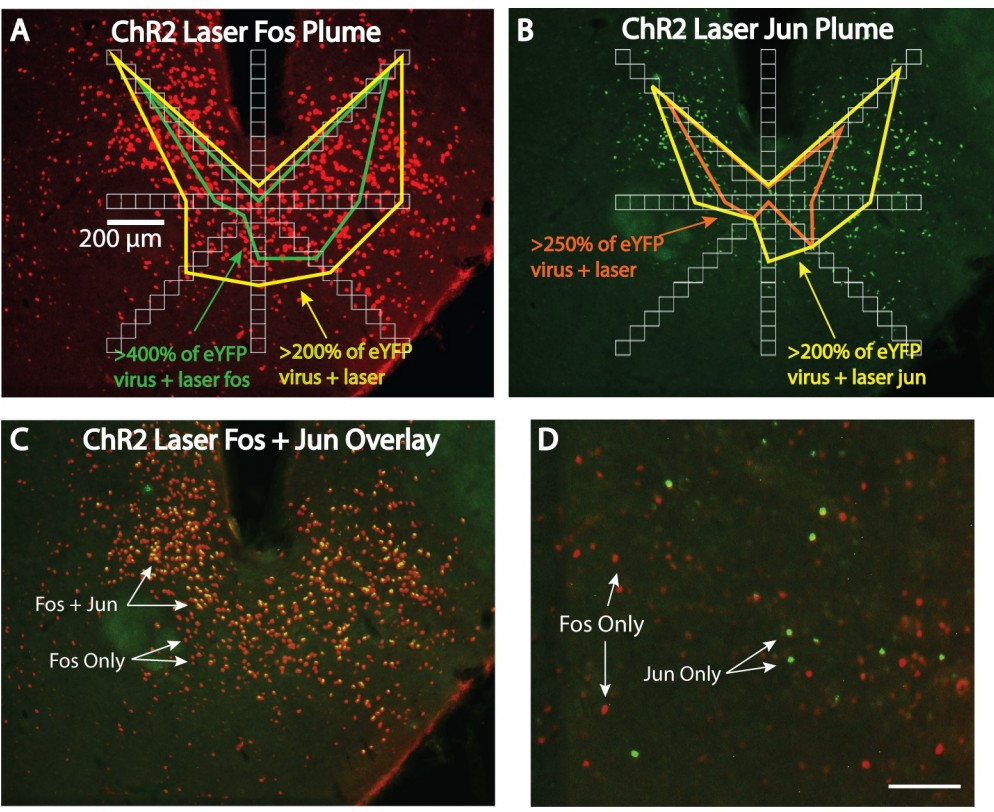

**Fig 3. Fos and Jun plume measurement process.** A: Fos plume from Fig 2 overlaid with an eight radial arm set of boxes for neuronal counting. Yellow and orange outlines represent elevation in neuronal Fos expression in ChR2 brains versus eYFP brains after both receive similar laser illumination. B: Jun plume from the same brain section. C: Overlap of Fos (red) and Jun (green) expression in the same brain. Optogenetic Jun plumes overlap almost exclusively within Fos plumes. D: Photomicrograph from the same brain section but in the dorsomedial hypothalamus, showing non-overlapping Fos and Jun expression. Scale bars represent 100 µm unless otherwise stated.

excites only the proportion of infected neurons that are close enough to receive light intensity exceeding neuronal illumination thresholds needed to trigger immediate early gene transcription and translation into protein.

Within a Fos plume, Fos immunoreactive nucleus "granules" were not only more numerous, but also showed brighter staining (determined by higher mean gray value in ImageJ) than granules located in Fos-expressing neurons outside the plume, measured in randomly-selected neighboring brain regions such as the arcuate nucleus, dorsomedial hypothalamus, or amygdala, (Fig 2E; significant effect of location: $F(1,51) = 16.44$, $p < 0.001$, $\eta^2 = 0.031$). Laser illumination was required to induce ChR2 brightness within the plume radius; there was a significant effect of group (ChR2 with laser vs ChR2 without laser vs eYFP with laser: $F(2,45) = 61.957$, $p < 0.001$, $\eta^2 = 0.548$), and illumination and virus type interacted significantly with location for granule brightness ($F(2,51) = 36.092$, $p < 0.001$, $\eta^2 = 0.136$). Further pairwise testing within this interaction revealed that Fos granules for neurons inside the plume were significantly brighter than stained granules in LH neurons outside the plume of elevated Fos expression in the ChR2 laser group ($M_{in} = 231.39 \pm 4.00$ vs $M_{out} = 177.96 \pm 6.45$, $t(17) = 10.251$, $p < 0.001$, $d = 2.35$; Fig 2E). Laser illumination of ChR2 was required for both the increase in the number of neurons expressing Fos and the increase in brightness for neurons that did express Fos; in the absence of laser, there was no difference in staining intensities of

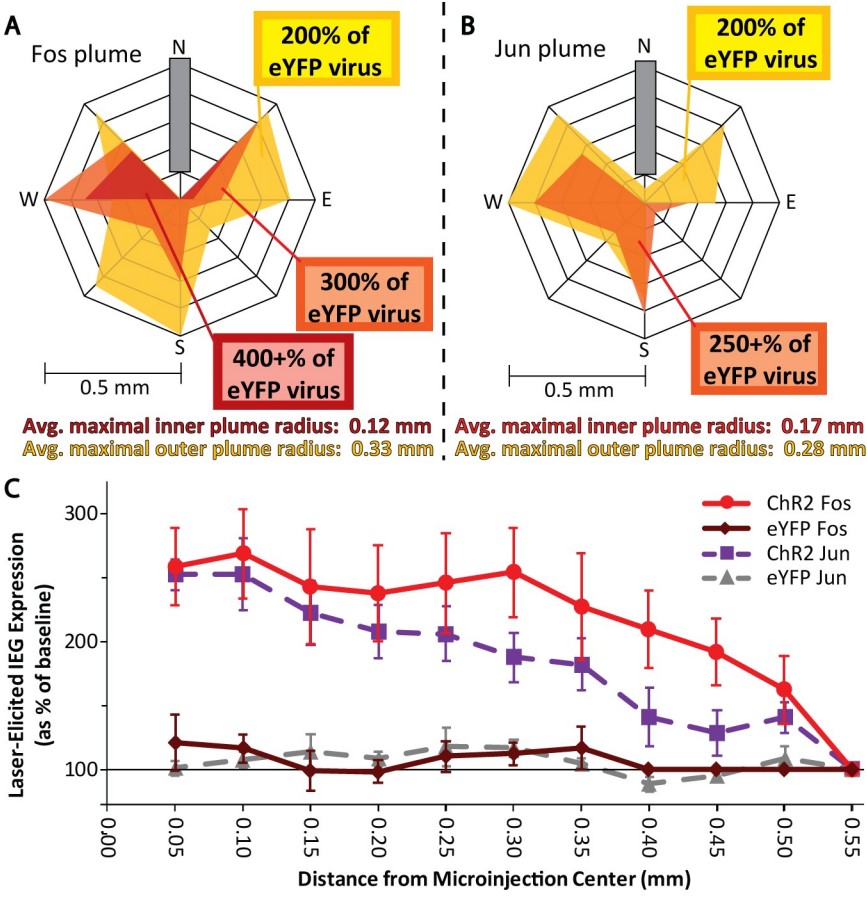

**Fig 4. Fos and Jun plume metrics.** A: Calculated average Fos plume. B: Calculated average Jun plume. C: Distance of elevated Fos and Jun expression relative to brains with inactive virus.

granules inside versus outside of the radius of a Fos plume for the ChR2 no laser group ($M_{in}$ = 134.4 ± 6.32 vs $M_{out}$ = 147.76 ± 5.06) or the eYFP laser group ($M_{in}$ = 149.85 ± 5.18 vs $M_{out}$ = 148.25 ± 4.54) (Fig 2E).

### Laser stimulation of specific LH subregions evokes food intake

**Initial food intake.** For LH overall, without taking subregional localization into account, stimulation of LH ChR2 rats produced an overall increase in food intake due to initial laser illumination ($F(1,20)$ = 14.23, p = 0.001, $\eta^2$ = 0.091). When subregion location along the anterior-posterior (AP) dimension was taken into account, there also were significant effects of both Laser ($F(1,16)$ = 17.922, p = 0.001, $\eta^2$ = 0.089) and Subregion ($F(4,16)$ = 8.450, p = 0.001, $\eta^2$ = 0.213), as well as a Laser x Subregion interaction ($F(4,16)$ = 3.908, p = 0.021, $\eta^2$ = 0.077), indicating localization of function effects on food intake.

Localization effects were mapped using individual site symbols sized to match our Fos/Jun plume metrics. These symbols were placed in a brain atlas location to match the histological localization of fiber tips for an individual rat, and colored to reflect the level of food intake measured for the same individual rat, thus showing laser-mediated food intake changes in various ChR2 rats mapped along three dimensions, corresponding AP, ML, and DV stereotaxic axes (Fig 5). When comparing laser-simulated intake increases between AP subregions, significant subregion effects emerged in laser-mediated intake ($F(4,16)$ = 4.592, p = 0.012, $\eta^2$ =

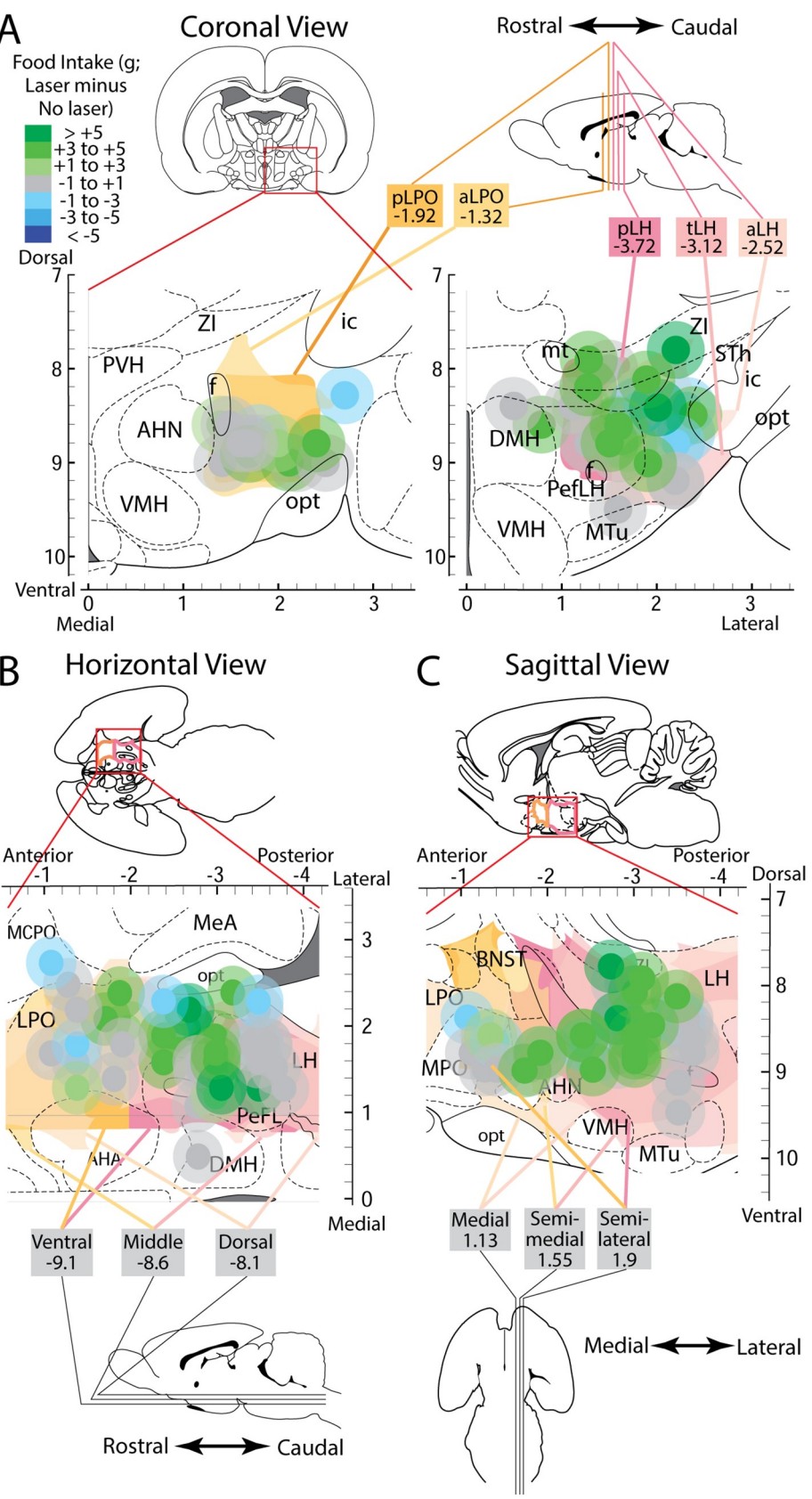

**Fig 5. Fos/Jun plumes and associated laser-mediated food intake intensities localized to stereotaxic atlas maps.**
Placements were localized in along the AP axis to the LPO (A, left) and LH (A, right), along the ML axis (B), and along
the DV axis (C). Different anatomical levels within these axes were topographically layered and colored to show levels
of the LPO in shades of orange and levels of LH in shades of pink. Plumes were colored to show magnitude of
behavioral laser-elicited food intake change versus baseline intake levels, with outer plumes made translucent.

0.534) and in the difference between laser intake minus non-laser intake for the same rat (F
(4,16) = 3.136, p = 0.044, $\eta^2$ = 0.439). Laser-induced change in intake was significantly greater
for rats that had sites in the tuberal LH (tLH) subregion than for sites in either the posterior
aLPO (p = 0.028) or pLH (p = 0.011) subregions (Fig 6A, top bar graph).

Intakes during laser periods versus non-laser periods were compared within each rat, and
analyses were performed both for the entire LH group as a whole, and for subgroups with sites
in specific LH or LPO subregions. Overall for all LPO and LH sites combined, laser illumina-
tion stimulated increases of nearly 300%, significantly above non-laser baseline intake of the

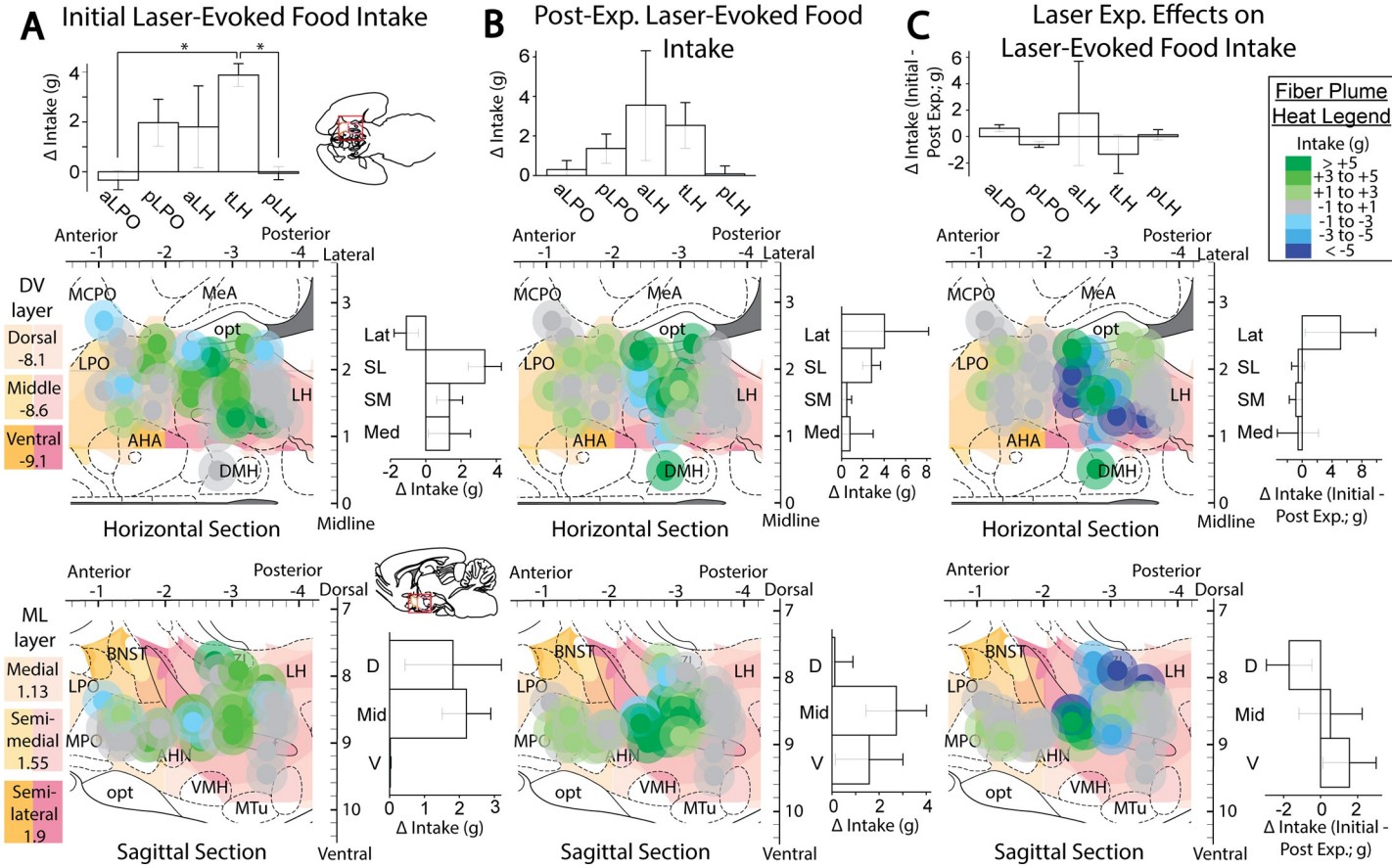

**Fig 6. Anatomical localization of laser-mediated food intake.** Fos plume placement maps demonstrating laser effects on food intake in ChR2 subjects prior to (A), after
(B), and magnitude of change due to (C) extended laser exposure. Plume locations are mapped onto horizontal and sagittal portions of stereotaxic atlas pages (29). Each
dot represents one subject's unilateral Fos plume scaled to estimated size of 200+% of baseline expression, and color represents magnitude of eating response (defined as
the individual's intake difference between laser and no laser conditions). Bilateral placements are both overlaid onto the same unilateral diagram. Bar graphs represent
intake difference scores (defined as group's average laser session intake minus average non-laser session intake, assessed separately for different anatomical levels). Bar
graphs depict change in intake in grams, calculated as intake under laser stimulation minus intake without laser stimulation in panels A and B, and then post-experience
change in intake minus pre-experience change in intake in panel C. Brain area abbreviations: AHA—anterior hypothalamic area; BNST–bed nucleus of stria terminalis;
DMH–dorsomedial hypothalamus, HDB–horizontal diagonal band of Broca; LH–lateral hypothalamus; LPO–lateral preoptic area; MCPO–magnocellular preoptic area;
MeA–medial amygdala; opt–optic tract; VP–ventral pallidum; ZI–zona incerta. Bar graph labels: Lat–lateral; SL–semi-lateral; SM–semi-medial; Med–medial; D–dorsal;
Mid–middle; V–ventral. *: p < 0.05, **: p < 0.01.

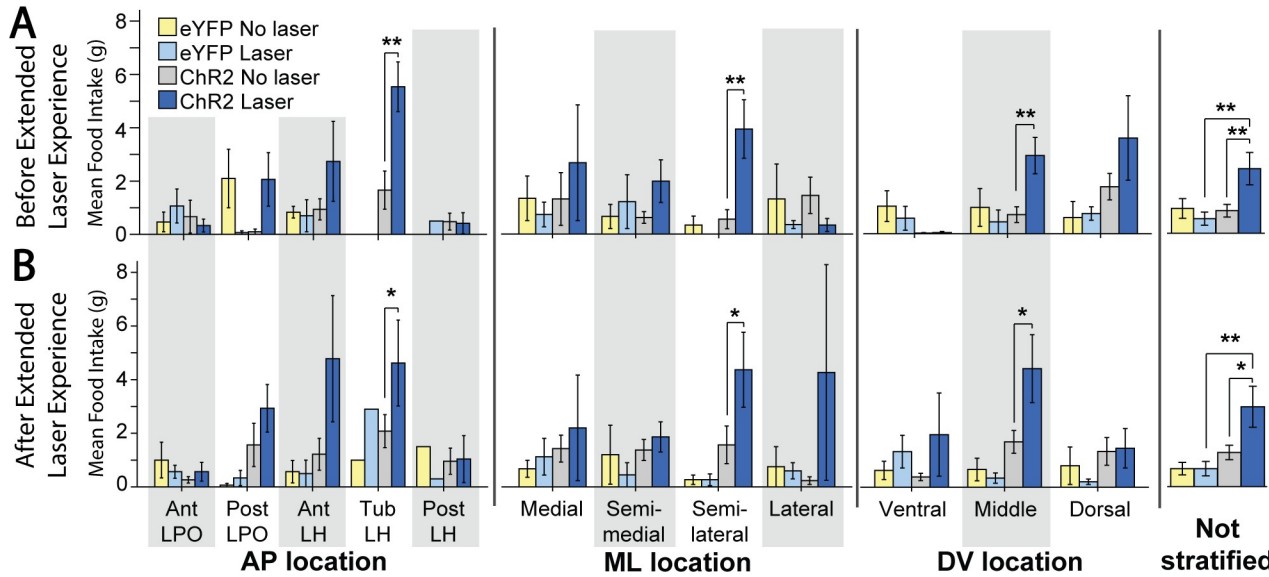

**Fig 7. Laser-elicited intake comparisons between groups.** Food intake for each virus group with or without laser, subdivided into AP, ML, or DV axis subgroups or for entire LH groups combined. A: Intake amounts prior to extended laser exposure. B. Intake amounts after extended laser exposure. *: p < 0.05, **: p < 0.01.

same rats ($M_L$ = 2.41 +/- 0.61 vs $M_{NL}$ = 0.84 +/- 0.24, t(20) = 2.973, p = 0.004 (one tailed), d = 0.74; Fig 7A, "Not stratified" column). However, among AP subregions, only in the tLH subgroup did laser stimulation produce significantly greater food intake, which was on average >300% above no-laser baseline periods of the same rats ($M_L$ = 5.54 +/- 0.93 vs $M_{NL}$ = 1.66 +/- 0.71, t(4) = 8.540, p < 0.001 (one-tailed), d = 2.09; Fig 7A, "AP location" column). Along the DV dimension, only in the middle location was laser-stimulated food intake significantly greater than non-laser food intake, an increase of approximately 400% over baseline ($M_L$ = 2.91 +/- 0.68 vs $M_{NL}$ = 0.71 +/- 0.29, t(9) = 3.159, p = 0.006 (one-tailed), d = 1.33; Fig 7A, "DV location" column) These results indicate that the tLH subregion of the AP axis, overlapping with the middle stratus of the DV axis, is where laser ChR2 stimulation most readily increased food intake.

Laser illumination did not alter intake in the eYFP control group with optically-inactive virus; this eYFP control group showed no significant intake differences overall, nor in any specific LH or LPO subregion. Control eYFP rats therefore had lower food intake overall than ChR2 rats during laser-on periods ($M_{ChR2}$ = 2.41 +/- 0.61 vs $M_{eYFP}$ = 0.55 +/- 0.24, Levene's p = 0.001, t(25.622) = 2.859, p = 0.004 (one tailed), d = 0.91; Fig 7A, "Not stratified" column), yet these groups did not significantly differ during laser-off periods ($M_{ChR2}$ = 0.84 +/- 0.24 vs $M_{eYFP}$ = 0.93 +/- 0.37, t(30) = -0.201, p = 0.421 (one tailed)). There were no significant interaction effects of Virus Group x Subregion. These results indicate that laser stimulation of food intake in LH requires ChR2 excitation of neurons, and is not simply due to laser light or heat, nor to AAV virus infection.

**Food intake after extended laser experience.** After 3 daily 2-hr sessions of extended exposures to laser illumination with food available, food intake was re-tested in the same rats as above. Overall, laser illumination in ChR2 rats in all combined LH and LPO subregions still increased food intake over their non-laser baseline periods to a similar degree as in initial tests ($M_L$ = 2.99 +/- 0.77 vs $M_{NL}$ = 1.28 +/- 0.27, t(20) = 2.298, p = 0.016 (one-tailed), d = 0.65; Fig 7B, "Not stratified" column). However, several individual rats that initially had shown

moderate to strong "eating-only" (> 3 g intake during laser), with sites in pLPO, aLH and tLH, switched to showing moderate self-stimulation and/or lower eating after extended exposure. Two other rats with sites in the aLH to tLH and middle DV subregion, showed increases in laser-bound eating of nearly an order of magnitude after extended laser experiences with food. In original tests, these rats failed to eat more when laser was on than when it was off, and actually ate slightly less during illumination periods (laser minus non-laser intake in grams; M = -1.15 +/- 1.15). After extended laser/food experience, in their later test session the same rats ate almost 10 g more food when laser was on than when laser was off (M = 9.75 +/- 2.55). Subtracting pre-experience intake from post-induction intake, these two rats showed a significant induction of increased stimulation-bound eating compared to other rats in the same AP subregions ($M_{strong}$ = 10.9 +/- 3.70 vs $M_{APnon}$ = -2.46 +/- 1.06, t(8) = 5.045, p = .001 (two-tailed)) and compared to others in the DV-middle region ($M_{strong}$ = 10.9 +/- 3.70 vs $M_{DVnon}$ = -1.03 +/- 0.78, t(9) = 5.446, p < .001 (two-tailed)). However, we note that the majority of rats with similar sites in aLH to tLH subregions failed to show any induction increase in intake after extended laser experience (N = 8; $M_{pre}$ = 3.84 +/- 0.69 vs $M_{post}$ = 1.38 +/- 1.05). That suggests that induction of potentiated stimulation-bound eating is not simply a reliable function of anatomical LH subregion. Rather, induction of eating via aLH/tLH laser experience may be a potent phenomenon for individuals in which it occurs, but is only shown by a small minority of individual rats (e.g., 20%) that undergo the extended laser-experience sessions.

## Stimulation of the tuberal LH subregion supports place-based laser self-stimulation, but only after extended laser experience

In the place-based laser self-stimulation test, significant effects on place preference were observed in the ChR2 group for AP subregion (F(4,15) = 3.837, p = 0.024, $\eta^2$ = 0.079) and Corner x AP subregion (F(4,15) = 3.846, p = 0.024, $\eta^2$ = 0.317). When accounting for both AP subregion and sex, significant effects were seen for Corner (F(1,10) = 8.304, p = 0.016, $\eta^2$ = 0.003), Extended laser experience x Sex (F(1,10) = 13.847, p = 0.004, $\eta^2$ = 0.006), Experience x Subregion (F(4,10) = 11.477, p = 0.001, $\eta^2$ = 0.019), Experience x Sex x Subregion (F(4,10) = 12.596, p = 0.001, $\eta^2$ = 0.021), Experience x Corner (F(1,10) = 8.207, p = 0.017, $\eta^2$ = 0.013), and Experience x Corner x Sex (F(1,10) = 13.673, p = 0.004, $\eta^2$ = 0.022). This pattern suggested a strong role for extended laser experience in shaping the ability of LH stimulation to support place-based self-stimulation.

Accordingly, initial place-based self-stimulation tests did not reveal a preference for the laser-delivering corner, nor an avoidance of it. However, significant differences in corner preference emerged after extended laser experience, which then differed across subregions (F (4,15) = 6.116, p = 0.004, $\eta^2$ = 0.620). Specifically, after extended laser experience, the tLH group showed the most pronounced preference for the laser-delivering corner ($M_{laser}$ = 54.0 +/- 8.7 vs $M_{other}$ = 15.3 +/- 2.9; t(4) = 3.350, p = 0.029, d = 2.67; Fig 9B, "AP location" column). This laser-corner preference of the tLH group was higher than that of most of the other subregions (aLPO group (p = 0.01), pLPO group (p = 0.018), aLH group (p = 0.011); Fig 8B, top graph), which did not exhibit significant preferences for the laser-corner. Comparing preferences of tLH rats before versus after extended laser exposure, a significant increase occurred over time in the magnitude of laser corner preference ($M_{post}$ = 54.0 +/- 8.7 vs $M_{pre}$ = 35.6 +/- 11.1, t(4) = 3.015, p = 0.039 (two tailed), d = 0.82; Fig 9B, "AP location" column, Tub LH cluster). Sex further modulated this experience-induced tLH stimulation preference, in that tLH females in particular had a greater increase in laser corner preference after extended experience than tLH males ($M_{male}$ = 9.0 +/- 3.8 vs $M_{female}$ = 32.5 +/- 0.4; t(3) = -4.764, p = 0.018 (two tailed), d = -5.02; Fig 9C, AP location, Tub LH cluster). The tLH females specifically showed a

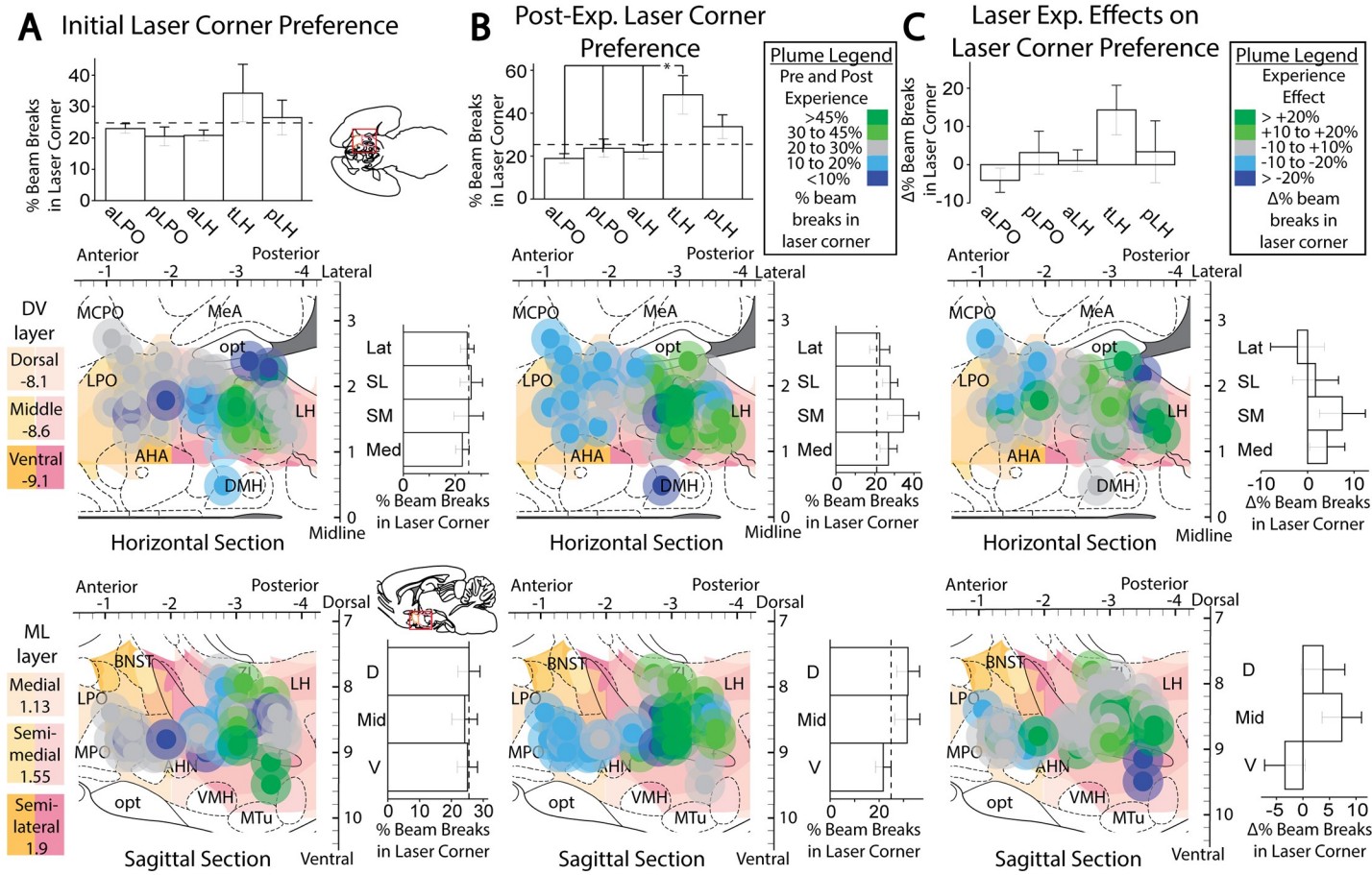

**Fig 8. Anatomical localization of place-based self-stimulation.** Fos plume maps demonstrating place-based laser self-administration in ChR2 subjects prior to (A), after (B), and due to (C) induction. Data depict percent of total entries or movements in laser-delivering corner versus other corners; dotted lines represent levels due to chance. Figure conventions and abbreviations are similar to previous mapping figures. Bar graphs represent laser corner entries for different anatomical subregions. *: p < 0.05, **: p < 0.01.

growth over time in laser-corner preference ($M_{post}$ = 45.2 +/- 7.8 vs. $M_{pre}$ = 12.7 +/- 8.1; t(1) = 92.714, p = 0.007 (two-tailed), d = -2.89), whereas tLH males did not.

By contrast, rats in the pLPO group showed an avoidance of their laser corner, which similarly emerged after extended experience ($M_{laser}$ = 18.1 +/- 5.3 vs $M_{other}$ = 27.3 +/- 1.8; t(2) = -9.074, p = 0.012, d = -1.36; Fig 9B, AP location column). Again, sex was a factor, as laser corner was avoided more strongly by pLPO females than by pLPO males ($M_{male}$ = 28.6 +/- 0 vs $M_{female}$ = 12.8 +/- 0.5; t(1) = 18.244, p = 0.035 (two tailed), d = 31.5; Fig 9B, AP location column). These data indicate that, especially after extended laser experience, tLH females exhibit an increased positive preference while pLPO females exhibit an increased negative avoidance of their respective laser-delivering corner. Control eYFP group rats did not exhibit any significant effects of laser, location, or any between-subjects factors on corner preference (Fig 9).

### Active self-stimulation in object-touch task is enhanced by extended laser experience

In the active-touch task, where voluntarily touching a designated empty metal spout object produced laser-illuminations (laser spout) while touching another empty spout produced nothing (control spout), we found significant self-stimulation effects in the overall combined

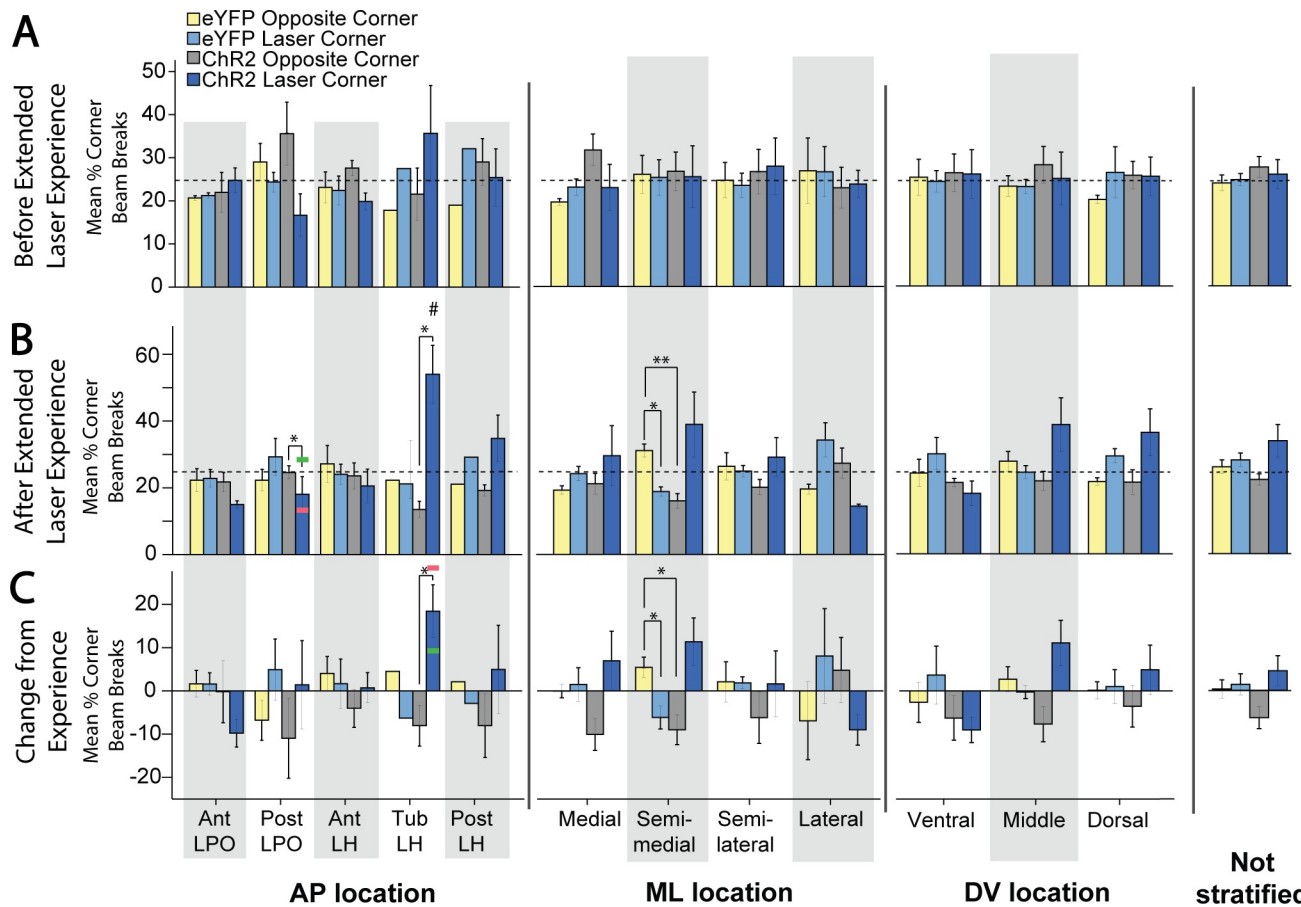

**Fig 9. Place-based laser self-administration comparison between groups.** These data depict percent of entries into laser-delivering corner compared to alternative corners, comparing ChR2 rats versus eYFP control rats. These data are subdivided into AP, ML, or DV axis levels, or presented for entire LH group combined. A: Corner preferences prior to extended laser exposure. B: Corner preferences after extended laser experience. C: Changes in corner preferences due to extended laser exposure. *: p < 0.05; **: p < 0.01; #: comparison of tLH ChR2 laser corner preference after vs before extended later experience, p < 0.05; green and pink horizontal bars represent male and female mean scores in that group.

LPO/LH ChR2 groups (Spout: $F(1,13) = 8.337$, p = 0.013, $\eta^2 = 0.195$), which was influenced by Extended Laser Experience ($F(1,13) = 8.674$, p = 0.011, $\eta^2 = 0.037$), causing an Experience x Spout interaction ($F(1,13) = 7.139$, p = 0.019, $\eta^2 = 0.03$). Additional LH subregional main and interaction effects also were found (Subregion: ($F(4,9) = 10.188$, p = 0.002, $\eta^2 = 0.317$; Spout: ($F(1,9) = 8.643$, p = 0.016, $\eta^2 = 0.085$); Spout x Subregion ($F(4,9) = 6.961$, p = 0.008, $\eta^2 = 0.275$)), Extended Experience ($F(1,9) = 10.162$, p = 0.011, $\eta^2 = 0.019$), Experience x Subregion ($F(4,9) = 6.666$, p = 0.009, $\eta^2 = 0.05$), Experience x Spout ($F(1,9) = 7.801$, p = 0.021, $\eta^2 = 0.012$), Experience x Spout x Subregion ($F(4,9) = 8.125$, p = 0.005, $\eta^2 = 0.052$).

**Initial active spout-touch self-stimulation.** Across all ChR2 rats in all LH and LPO subregions combined, LH/LPO illumination supported active self-stimulation even on initial tests, as ChR2 rats made >300% more touches on the laser spout that triggered 1-sec laser illuminations than on their control non-laser spout ($M_L = 79.1 +/- 27.0$ vs $M_{NL} = 19.7 +/- 3.3$, t (20) = 2.225, p = 0.019 (one tailed), d = 0.67; Fig 10A, "Not stratified" column), and similarly made 300% more touches on the laser spout than eYFP control rats did ($M_{ChR2} = 79.1 +/- 27.0$ vs $M_{eYFP} = 23.2 +/- 3.3$, Levene's p = 0.006, t(20.588) = 2.058, p = 0.026 (one tailed), d = 0.64). Thus, the difference between laser-spout vs non-laser spout was also greater for ChR2 rats

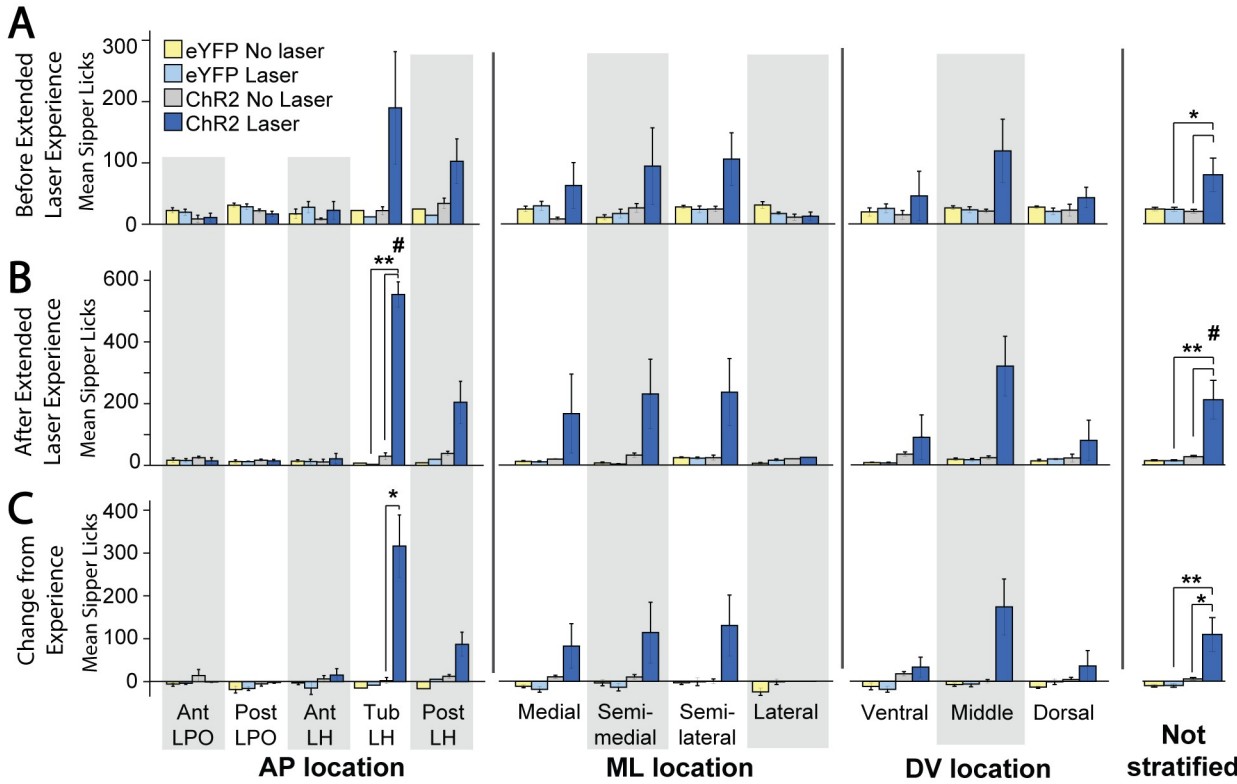

**Fig 10. Active laser self-administration on spout-touch task, compared between groups.** These ChR2 versus eYFP data are subdivided into AP, ML, or DV axis subgroups, or presented for entire LH group combined. A: Spout contacts prior to extended laser exposure. B. Spout contacts after extended laser exposure. C: Changes in spout contacts due to extended laser exposure. *: p < 0.05; **: p < 0.01; #: comparison of ChR2 laser corner preference after vs before extended later experience, p < 0.05.

than eYFP rats ($M_{ChR2}$ = 59.3 +/- 26.7 vs $M_{eYFP}$ = -0.5 +/- 4.7, Levene's p = 0.010, t(21.204) = 2.210, p = 0.019 (one-tailed), d = 0.69). Thus, consistent with decades of electrode stimulation studies, excitation of LH/LPO neurons generally is sufficient in naïve rats to support self-stimulation requiring active instrumental responses.

**Extended laser exposure further enhances active self-stimulation.** Extended experience with laser stimulation, although not required for initial self-stimulation on the active spout-touch task, still exerted a facilitatory effect on instrumental ChR2 self-stimulation, which grew in magnitude from initially about 100 illuminations to subsequently about 200 illuminations per session after further laser experience ($M_{PostL}$ = 209.8 +/- 62.2 vs $M_{PreL}$ = 100.7 +/- 36.2, t (14) = 2.770, p = 0.015 (two tailed), d = 0.55; Fig 10B, "Not stratified" column). Specifically, for all LH/LPO groups combined, extended laser experience roughly doubled the number of ChR2 laser spout contacts but not control spout contacts ($M_{L\_Diff}$ = 109.1 +/- 39.4 vs $M_{NL\_Diff}$ = 5.5 +/- 3.3, t(14) = 2.631, p = 0.020 (two tailed), d = 0.96). Female and male ChR2 rats did not differ in their levels or patterns of spout-touch self-stimulation. Consequently, laser spout contacts remained much greater than non-laser contacts ($M_L$ = 209.8 +/- 62.2 vs $M_{NL}$ = 26.4 +/- 4.2, t(14) = 2.225, p = 0.005 (one tailed), d = 1.08).

Anatomical differences between LH subregion groups emerged after extended laser experience. AP differences between tLH, aLH and pLH groups influenced the final magnitude of self-stimulation (F(4,10) = 23.843, p < 0.001, $\eta^2$ = 0.905), the difference between control and laser spout contacts (F(4,10) = 20.021, p < 0.001, $\eta^2$ = 0.889), and the growth of self-

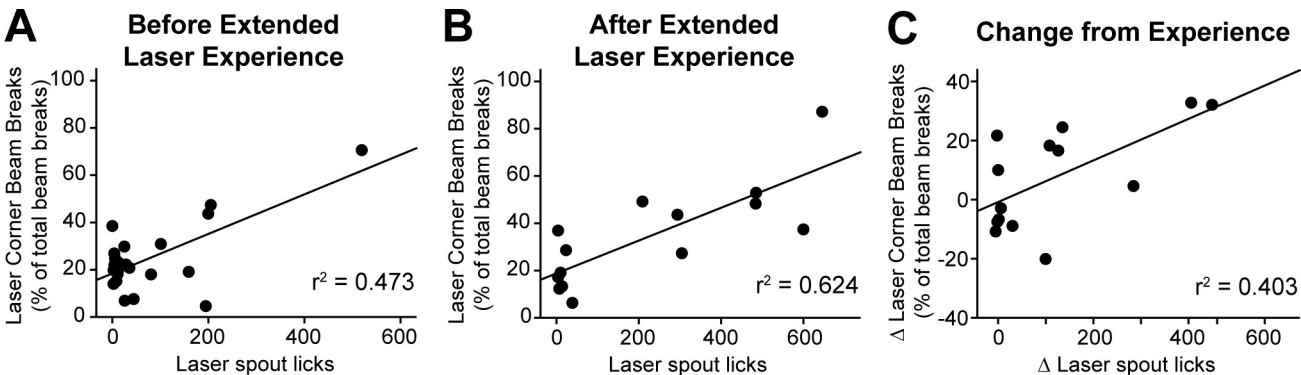

**Fig 11. Anatomical localization of spout-touch self-stimulation.** Fos plume maps demonstrating spout-touch laser self-administration in ChR2 rats prior to (A), after (B), and magnitude of change due to (C) extended laser exposure. Data depicts spout touch score–the difference of laser-associated spout touches minus alternative non-laser spout touches. Figure conventions and abbreviations are the same as the prior mapping figures. Bar graphs represent spout-touch difference scores (average laser licks minus average non-laser licks) for each LH site. *: p < 0.05, **: p < 0.01.

stimulation from before versus after laser experience (F(4,10) = 8.681, p = 0.003, $\eta^2$ = 0.776). In particular, tLH subregion group reached the highest final level of self-stimulation of 350–500 illuminations per session greater than the other four subregions: tLH self-stimulation was

**Fig 12. Relationship between the outcomes of two laser self-stimulation tasks.** Correlation plots showing the relationship between place-based laser self-administration and active spout-touch laser self-administration, before (A), after (B), and change due to (C) extended laser exposure.

orders of magnitude higher than either the aLPO group (p < 0.001), pLPO group (p < 0.001), aLH group (p < 0.001), or pLH group (p = 0.002). Similarly, the difference in laser versus control spout contacts was highest in the tLH, higher than in the aLPO (p < 0.001), pLPO (p < 0.001), aLH (p = 0.001) or pLH (p = 0.002) (Fig 11B, top graph). The relative growth in self-stimulation induced by extended laser experience over initial levels was an increase of 217 stimulations in the tLH, far greater than the decreases of 1.6 stimulations in the aLPO (p = 0.011) and 2.1 stimulations in the pLPO (p = 0.004), and increases of 15 stimulations in the aLH (p = 0.015) and 87 stimulations in the pLH (p = 0.023) (Fig 11C, top graph). Lastly, the tLH group significantly increased their laser spout contacts after extended laser experience ($M_{PostL}$ = 553.7 +/- 41 vs $M_{PreL}$ = 237.3 +/- 101.5, t(3) = 4.374, p = 0.022 (two tailed), d = 2.04; Fig 10B, "AP location" column). Overall, the tLH appeared to be the anatomical subregion that grew with extended experience to highest levels of laser self-stimulation by active spout touching, just as the tLH had grown to highest self-stimulation in the more passive place-based self-administration task.

Control eYFP rats failed to self-stimulate at any LH site, and were consequently much lower than ChR2 rats in laser spout contacts ($M_{ChR2}$ = 209.9 +/- 62.2 vs $M_{eYFP}$ = 13.3 +/- 2.6, Levene's p < 0.001, t(14.048) = 3.159, p = 0.004 (one tailed), d = 1.15; Fig 10B, "Not stratified" column). No LH or LPO site supported laser self-stimulation in eYFP subjects.

## Laser place preference correlates with self-stimulation and both increase after extended laser experience

Comparing an individual rat's self-stimulation in the active-touch task to its self-stimulation in the place-based task (Fig 12) a positive correlation was found between the scores across individuals, but only after extended laser exposure (r(12) = 0.79, p < 0.001). Prior to laser exposure, one strong responder drove this effect (Fig 12A); after omitting this outlier, the correlation was not significant (r(19) = 0.275, p = 0.241). The magnitude of increase induced by extended laser exposure was also correlated across spout-touch and place-based-preference self-stimulation measures (r(12) = 0.634, p = 0.015; Fig 12B). Thus, both measures of self-stimulation seem to be tapping into the same incentive motivation process related to response reinforcement.

By contrast, stimulation-bound increases in food intake did not correlate with either measure of laser self-stimulation (spout-touch correlation with intake: r(23) = 0.291, p = 0.159; corner preference correlation with intake: r(22) = 0.374, p = 0.074). Even after extended laser experience, food intake failed to correlate with either spout touches (r(12) = 0.161, p = 0.583) or place-based self-stimulation (r(20) = -0.045, p = 0.844). Thus, the ability of LH optogenetic stimulation to cause increases in eating behavior seem separable from its ability to support laser self-stimulation.

To further explore the relation between laser effects on food intake versus self-stimulation, rats with laser-induced increases in food intake greater than 2.5 grams of non-laser baseline were labeled as strong intake responders. Rats with increases in laser spout self-stimulation greater than 150 illuminations were labeled as strong self-stimulators (Fig 13). Extended laser experience did not significantly alter the total number of strong responders of any type according to pairwise Wilcoxon testing. However, anatomical trends in the incidence of strong responders were observed between AP groups for both Strong Eaters ($\chi^2$(4) = 8.571, p = .067, $\varepsilon^2$ = 0.429) and for Strong Self-Stimulators (Before experience: $\chi^2$(4) = 7.4, p = .072, $\varepsilon^2$ = 0.37; After experience: $\chi^2$(4) = 11.429, p = .014, $\varepsilon^2$ = 0.571). More strong eaters were found for the tLH than for either the aLPO group ($Rank_{aLPO}$ = 2.5 vs $Rank_{tLH}$ = 5.7; U = 1.5, Z = 2.049, p = 0.04, $r^2$ = 0.525) or the pLH group ($Rank_{pLH}$ = 3.5 vs $Rank_{tLH}$ = 7.5; U = 2.5, Z = 2.449,

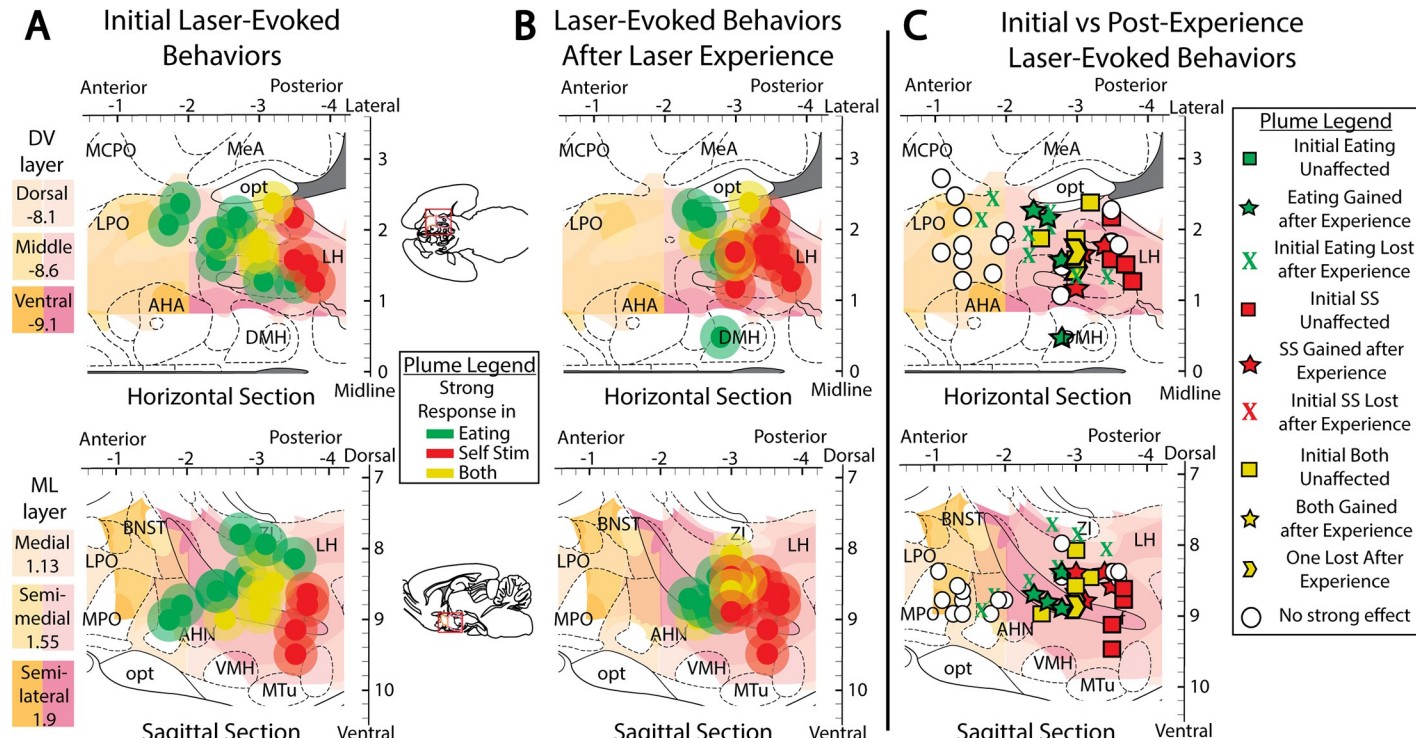

**Fig 13. Extended laser experience alters distribution of strong eaters vs strong self-stimulators.** Fos plume maps showing strong laser-bound eating (> 3 g) and/or strong laser self-administration (> 200 laser spout touches) in ChR2 subjects before (A) and after (B) extended laser exposure, as well as how extended laser exposure altered behaviors in each rat mapped here (C). Scores were obtained by subtracting non-laser from laser results for each experiment type. Map conventions are the same as in prior figures.

p = 0.014, $r^2$ = 0.6). Similarly, the tLH group also had more Strong Self-Stimulators than the aLH group initially (Rank$_{aLH}$ = 4 vs Rank$_{tLH}$ = 7; U = 5, Z = 1.964, p = 0.05, $r^2$ = 0.386; Fig 13A), and also more than either the aLH group (Rank$_{aLH}$ = 1.5 vs Rank$_{tLH}$ = 4.5; U = 0, Z = 2.226, p = 0.025, $r^2$ = 0.833), the aLPO group (Rank$_{aLPO}$ = 1 vs Rank$_{tLH}$ = 3.5; U = 0, Z = 2, p = 0.046, $r^2$ = 0.8), or the pLPO (Rank$_{pLPO}$ = 2 vs Rank$_{tLH}$ = 5.5; U = 0, Z = 2.449, p = 0.014, $r^2$ = 0.857) after extended laser experience (Fig 13B & 13C).

## Laser stimulation evokes different behaviors at different power outputs

Different intensities of laser stimulation were compared in the food intake situation across 0.2, 1, 5, and 10 mW levels. For the ChR2 group, 0.2 mW laser merely increased locomotion, measured as cage crossing (F(2,22) = 4.349, p = .026, $\eta^2$ = 0.235; laser > after, p = .009), and reduced periods of immobility and doing nothing, compared to no-laser periods (Fig 14A; F(2,22) = 6.626, p = .006, $\eta^2$ = 0.210; laser < after, p = .015 and laser < interim, p = .007); this effect on immobility was mostly on the middle DV group (Interaction of DV group x period: F(4,18) = 5.312, p = .005, $\eta^2$ = 0.216; laser < interim: M$_{laser}$ = 5.2 +/- 1.3 vs M$_{interim}$ = 12.1 +/- 1.0; t(6) = -3.736, p = .01, d = -2.31). Laser output at 1.0 mW similarly produced a trend toward reduced periods of inactivity (F(2,22) = 3.257, p = .058, $\eta^2$ = 0.179; laser > interim, p = .045). Comparing ChR2 rats to eYFP control rats, 0.2 mW laser on ChR2 rats similarly produced more cage crossing (Fig 15B; M$_{ChR2}$ = 1.2 +/- 0.8 vs M$_{eYFP}$ = -0.4 +/- 0.4, t(21) = 1.721, p = .050, d = 0.71), more rearing (M$_{ChR2}$ = 1.7 +/- 1.1 vs M$_{eYFP}$ = -0.8 +/- 0.6, Levene's p = .037, t(16.891) = 1.993, p = .032, g = 0.78), and less time doing nothing than eYFP rats (Fig 15D;

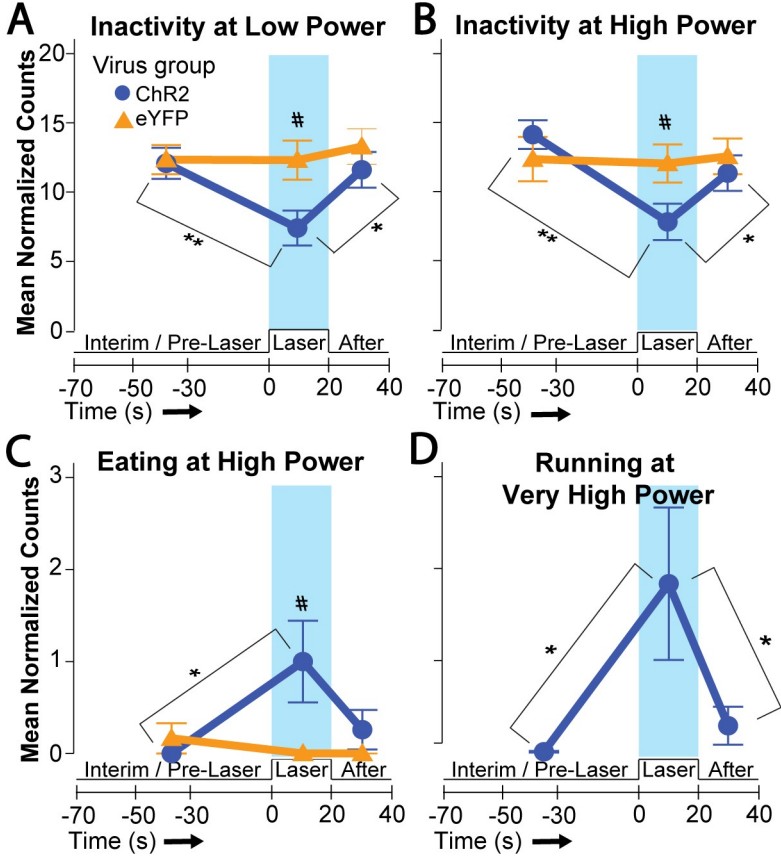

**Fig 14. Different LH laser intensities evoke different behavioral effects.** Line graphs show behavior prior to, during, and immediately after laser stimulation at designated laser power intensities. A and B: 0.2 and 5 mW laser power effects on inactivity. C: 5 mW laser stimulates increases in eating. D: 10mW laser stimulation increases running. Brackets with asterisks denote differences between laser periods versus pre/post no-laser periods for the ChR2 subject group; *: $p < 0.05$, **: $p < 0.01$. Pound sign denotes a significant difference between ChR2 vs eYFP groups; $p < 0.05$.

$M_{ChR2}$ = -4.7 +/- 1.4 vs $M_{eYFP}$ = 0 +/- 0.8, t(21) = -2.737, p = .006, d = -1.17). An interaction of sex with ChR2/eYFP virus was also found at 0.2 mW (F(1, 19) = 9.113, p = .007, $\eta^2$ = 0.223) such that female ChR2 subjects exhibited more cage crossing than eYFP females ($M_{ChR2}$ = 3.1 +/- 0.8 vs $M_{eYFP}$ = -0.4 +/- 0.6, t(10) = 3.184, p = .005, d = 2.00; Fig 15B, pink symbols). At 1.0 mW, comparing ChR2 rats to eYFP rats, rearing again was elevated in ChR2 subjects over eYFP levels ($M_{ChR2}$ = 1.8 +/- 1.1 vs $M_{eYFP}$ = -1 +/- 0.8, t(21) = 1.971, p = .031, d = 0.83).

Eating behavior became significantly increased by 5 mW laser compared to control periods of no-laser in the same ChR2 rats (Fig 14C; F(2, 36) = 3.903, p = .029, $\eta^2$ = 0.108; laser > interim, p = .037), and 5 mW also similarly decreased time doing nothing (Fig 14B; F(2,36) = 14.239, p < .001, $\eta^2$ = 0.201; laser < after and laser < interim, p = .013 and p < .001). Compared to eYFP control rats, 5 mW stimulation on ChR2 rats caused greater eating (Fig 14A; $M_{ChR2}$ = 1.0 +/- 0.4 vs $M_{eYFP}$ = -0.3 +/- 0.2, Levene's p = .029, t(24.547) = 2.666, p = .007, g = 0.80), more chow carrying ($M_{ChR2}$ = 0.1 +/- 0.07 vs $M_{eYFP}$ = -0.06 +/- 0.04, t(28) = 2.082, p = .024, d = 0.83), more running (Fig 15C; $M_{ChR2}$ = 0.8 +/- 0.4 vs $M_{eYFP}$ = 0 +/- 0, Levene's p = .011, t(18) = 1.849, p = .041, g = 2.43), and less time doing nothing (Fig 15D; $M_{ChR2}$ = -6.3 +/- 1.3 vs $M_{eYFP}$ = -0.3 +/- 0.9, Levene's p = .022, t(27.98) = -3.789, p < .001, g = -1.20). In addition, comparing anatomical subregions, the medial ML axis group also exhibited greater chow-carrying than the other ML groups (F(3,15) = 12.484, p < .001, $\eta^2$ = 0.713; medial > all

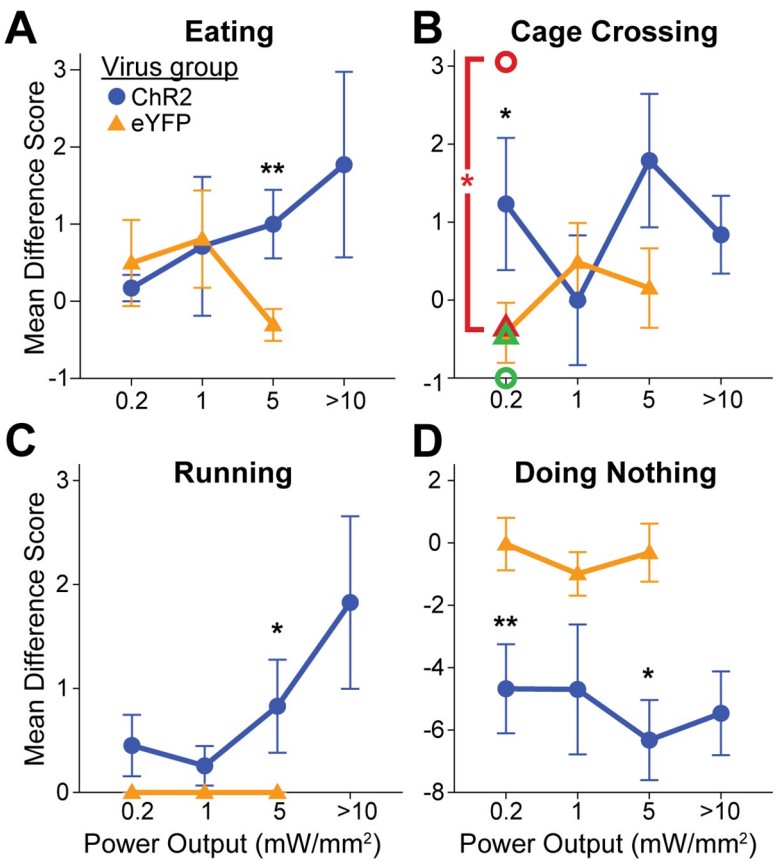

**Fig 15. Other behavioral effects of different LH laser intensities.** Line graphs depicting difference scores–derived by subtracting interim period counts from laser period counts–for eYFP and ChR2 groups at increasing laser power outputs. The behaviors depicted are eating (A), cage crossing (B), running (C), and doing nothing (D). A black asterisk denotes significant difference between eYFP and ChR2 groups at that laser power, whereas a pink asterisk denotes significant difference between eYFP female and ChR2 female groups; *: p < 0.05, **: p < 0.01. Pink and green symbols represent the means for female and male rats, respectively.

other groups, p < .001). For with ML group, there was also an interaction with ChR2/eYFP virus condition on chow carrying (F(3,22) = 9.192, p < .001, $\eta^2$ = .308) such that medial ChR2 rats carried chow across the cage more often than the medial eYFP subjects ($M_{ChR2}$ = 0.8 +/- 0.1 vs $M_{eYFP}$ = -0.1 +/- 0.1, t(4) = 6.339, p = .002, d = 5.45).

Lastly, at 10 mW laser power, time spent running increased (Fig 14D; F(2,32) = 5.063, p = .012, $\eta^2$ = 0.142; laser > after and laser > interim, p = .029 and p = .043) while doing nothing decreased (F(2,32) = 9.222, p = .001, $\eta^2$ = 0.168; laser < after and laser < interim, p = .007 and p = .001). Additionally, a number of anatomical differences emerged at 10 mW. For example, the pLH group exhibited the most running (F(4,12) = 8.899, p = .001, $\eta^2$ = 0.749; pLH > all other groups, p < .001), while the medial ML group showed the most toy gnawing (F(3,16) = 3.911, p = .034, $\eta^2$ = 0.476; medial > all others, p < .02), the lateral ML group showed the most cage crossing (F(3,16) = 5.042, p = .016, $\eta^2$ = 0.537; lateral > all others, p < .025), and the dorsal group of the DV axis trended toward more digging (F(2,14) = 3.688, p = .052, $\eta^2$ = 0.344; dorsal > middle, p = .019) and further showed more defensive treading behavior (F(2,14) = 3.877, p = .046, $\eta^2$ = 0.268; dorsal > others, p < .035). Furthermore, laser stimulation of some groups caused behavioral differences from their no-laser interim periods; the pLH AP group increased running (Interaction of AP group x period: F(8,24) = 9.184, p < .001, $\eta^2$ = 0.331;

laser > interim, $M_{laser}$ = 8.0 +/- 2.5 vs $M_{interim}$ = 0 +/- 0, p = .042, d = 2.63), and the lateral ML group increased cage crossing (Interaction of ML group x period: F(6,26) = 2.491, p = .049, $\eta^2$ = 0.22; laser > interim, $M_{laser}$ = 4.0 +/- 0.6 vs $M_{interim}$ = 0.3 +/- 0.1, t(2) = 8.517, p = .014, d = 5.01). Overall, inactivity decreased after application of laser at multiple outputs, eating was best stimulated specifically at 5 mW, and a variety of other behaviors emerged at 10 mW.

Comparing sexes within the ChR2 group, at 0.2 mW, females cage-crossed more than males (S1A Fig; $M_{male}$ = -1.3 +/- 0.8 vs $M_{female}$ = 3.1 +/- 0.8; t(10) = -3.814, p = .003, d = -2.30). At 1 mW, males groomed more than females (S1B Fig; $M_{male}$ = 1.2 +/- 0.8 vs $M_{female}$ = -0.9 +/- 0.3; Levene's p = .002, t(5.39) = 2.583, p = .046, g = 1.69). No other sex differences were significant. Control eYFP rats, by contrast, exhibited no differences between laser and non-laser periods at any outputs.

## Discussion

Here we found that optogenetic stimulation of LH neurons in different anatomical subregions may elicit distinctive patterns of behavior, especially regarding stimulation-bound eating and laser self-stimulation. By using laser-induced Fos plumes and Jun plumes in the LH as measures of the diameters of how far local excitation spread surrounding an illuminated optic fiber tip [27,32], we were able to precisely identify the subregion and size of tissue that became excited enough to alter immediate early gene expression. Increased c-fos gene transcription into mRNA and subsequently translation into Fos protein is a commonly-used genomic marker of neuronal activation, thought to reflect calcium influx associated with frequent firing [33], and c-jun gene translation of Jun protein provides an independent marker of neuronal immediate early gene activation. Both Fos plume and Jun plume diameters indicated that optogenetic excitation increased gene translation above 200% within a 0.3 mm radius from an LH fiber tip, compared to control baseline levels (measured at equivalent positions in laser-illuminated eYFP brains. Those plumes also contained smaller inner plumes of more intense >250% Fos/Jun elevations within a 0.15 mm radius. The similarity in sizes of Fos plumes and Jun plumes, and similarity in their percent elevation above control baselines, gives some confidence that each accurately reflects the size of the zone of direct neuronal excitation. There were some slight differences, however, as we modified the immunohistochemical procedure for staining Fos here via tyramide signal amplification [34], similar to chromogenic (DAB) staining methods, to more sensitively detect neuronal activation. This allowed realization that not only were more ChR2-expressing neurons also expressing Fos granules within laser-stimulated plumes compared to those neurons outside the plume boundaries, but also that Fos granules inside individual neurons within an illuminated plume stained with higher intensity than granules in other neurons outside the plume. Thus, both neuronal granular Fos intensity and the number of neurons expressing detectable Fos granules at all appear to be increased by direct optogenetic stimulation. The neuronal granule intensity measure may also distinguish between local direct optogenetic stimulation versus recruitment of neurons in neighboring or distant brain structures due to functional connectivity, which did not increase granule intensity.

### Anatomical localization of functions in LH subregions

Localization of function maps were created by mapping the behavioral effects in each individual rat onto its confirmed stereotactic site using plume-sized symbols. These function maps indicated that optogenetic stimulation of neurons specifically in the tuberal LH subregion may most potently elicit both laser-bound eating behavior and food intake, and support voluntary

laser self-stimulation behavior. This tuberal LH subregion is a roughly 0.36 mm$^3$ subregion located 2.81 mm to 3.4 mm posterior to bregma within the rostrocaudal axis, and intermediate within the dorsoventral axis. Collectively, our findings especially implicate the tuberal LH in laser-bound eating and in laser self-stimulation but also suggest that less intense motivational effects of ChR2 stimulation may extend throughout the LH, especially with additional laser experience. By contrast, laser avoidance and some escape-related behavior (mostly laser-bound running) was found at some LH sites, especially for posterior LH sites at highest mW illumination power. Lastly, some sex differences did emerge: ChR2 pLPO females showed laser place avoidance while ChR2 tLH females showed magnified laser place preference while males did not in both cases, ChR2 females showed more laser-bound cage crossing than other groups, and ChR2 males showed more laser-bound grooming than ChR2 females. The increased laser-bound locomotion effects in ChR2 female rats may be a magnification of their propensity to locomote more than male rats. The nature of the other sex differences seen here are less clear and warrant further study.

Activation of the tuberal LH also occurs in natural feeding and certain reward contexts. When eating patterns are entrained, LH Fos expression increases prior to a meal, especially when food-restricted [35]. Some of these activated neurons are orexin neurons known to primarily reside in the tuberal LH [36]. The LH is also a locus in which glutamate levels increase prior to and during the early parts of eating [37], and electrophysiological evidence shows that LH neurons activate to the sight of food-related cues and during hunger states [38,39]. Additionally, LH orexin neurons demonstrate increased Fos expression in drug and food reward-related contexts [40], and the LH shows increased Fos expression following VTA self-stimulation [41]. Lastly, electrical activity of LH neurons increases in response to sweet tastes [42]. In general, we think that optogenetically-effective subregions (e.g., tuberal LH) are included among brain regions showing IEG activation in natural feeding. However, IEG studies show correlation, and many of the activated structures may activate as consequences of feeding or as parallel mechanisms of other related functions, rather than directly cause feeding behavior. The optogenetic localization of function may more precisely identify specific causation of feeding behavior.

In our study, it also seems to be the case that certain individuals were reliable laser-bound eaters, while others are not, even for the same LH subregion. That is, while particular LH subregions have a much higher probability of supporting laser-induced eating than other subregions, the probability is not 100% across all individuals. We are trying to recognize such individual differences, in hope that future research may help reveal what individual features beyond LH subregion contribute to controlling stimulation-bound eating.

## Extended laser experience may modulate effects of LH stimulation

Extended experience with laser stimulation in the presence of food induced several changes in the behavioral effects of LH optogenetic stimulation. Most apparent was an overall increase in LH laser self-stimulation after extended laser experience, suggesting that the incentive value of optogenetic LH stimulation had strengthened either by mere repetition or by association with food. Overall, extended exposure did not strengthen stimulus-bound eating in our hands for the LH group as a whole nor for any anatomical subregion, in contrast to an early report of electrical LH stimulation [26]. However, more consistent with that report, we note that a few initially non-eating rats with sites in aLH and tLH changed into strong laser-bound eaters after extended laser experience, more similar to the induction of LH stimulation-bound eating or drinking after extended stimulation experience in the presence of one target reported by Cox and Valenstein. This appears to apply only to a minority of individuals under optogenetic stimulation experience for reasons that are not yet clear.

More generally, our results are consistent with the notion that extended laser experience can modulate the behaviors evoked by optogenetic stimulation at a given individual's site, and in our hands tended mostly to promote laser self-stimulation of LH sites. This may have implications for other optogenetic stimulation studies that assign particular functions to a neural substrate based on effects of relatively limited exposures to stimulation. It may be a mistake to assume that functional effects of optogenetic stimulation are intrinsic to the particular stimulated neural system, and likely to be stable and permanent. Our results suggest that LH-related circuitry may have functional flexibility, and that not all functions elicited by optogenetic stimulation are necessarily permanent or intrinsic to the stimulated neurons.

A potential explanation of laser-bound eating and laser self-stimulation induced by LH neuronal stimulation is that both behavioral effects reflect incentive salience activated by LH stimulation, which enhances the attractiveness and salience of potential food stimuli as well as enhancing the attractiveness of actions and stimuli paired with ChR2 stimulation, which then produces 'wanting' to self-stimulate laser activation. Such explanations have previously been offered to account for the motivating effects of LH electrodes [43][17], and seem equally applicable to effects of optogenetic LH stimulation here. However, our rats tended to show either laser self-stimulation or stimulation-bound eating here, but not both, at any one time. That was true even for rats that switched between the two after extended laser experience. One possibility is that incentive salience evoked by LH optogenetic stimulation at a given site/intensity becomes individually focused or channeled preferentially to either food as target, or to external spout/place stimuli that support self-stimulation. While the dominant target of incentive salience attribution may differ across individuals, or even perhaps change over time within the same individual, it may be that only one target typically predominates at a time. Thus, both electrical and optogenetic stimulation of LH subregions may increase incentive salience of stimuli, but there may be individual differences in the relative "weight" given to food stimuli versus self-stimulation stimuli (e.g., laser spout, laser corner), or even dynamic changes in target over time within the same individual.

## Anatomical features of LH subregions and localization of function

A recent optogenetic investigation of LH to VTA connections reported that a majority of LH fibers regulating both feeding and self-stimulation within the VTA likely originated from the tuberal LH subregion, around -2.8 mm posterior of bregma [44]. It is likely our manipulations of the same tuberal LH also operated through LH to VTA projections. However, we still obtained primarily feeding in the anterior LH and primarily self-stimulation in the semi-posterior LH. In contrast, Gigante et al. observed eating responses in only 3 of 17 aLH to VTA projection rats and self-stimulation in only 4 of 17 pLH to VTA projection rats.

Our identification of the tuberal LH as an especially potent subregion for eliciting eating seems similar to previous studies that implicated the same LH subregion in eating elicited by drug microinjections that act on AMPA, NMDA or kainate glutamate receptors, or on GABA-A receptors [45–48]. However, microinjections may occupy a significant volume of brain tissue, and without Fos plumes or Jun plumes it is difficult to know how far a drug microinjection spreads to alter neuronal function, even with fluorescently-conjugated drugs [49]. Here, the use of Fos plumes and Jun plumes gives confidence in stating that tLH optogenetic stimulations were contained essentially within the tuberal LH region. Definition of ChR2 plumes as elevations above those produced by laser illumination in control eYFP brains helps minimize the risk of confounds induced by local heat or light effects on neuronal function that are independent of ChR2-induced depolarization [50]. We also used relatively low power outputs, 2–5 mW/mm$^2$, to evoke eating and laser-self-stimulation, compared to 10–20 mW/mm$^2$

used in some other LH studies [51,52]. Lower laser power seems likely to produce smaller regions of activation than higher illumination intensities, and stimulating smaller regions with intensities that are still behaviorally effective may also help precise localization of function for optogenetic effects.

Among LH subdivisions, the aLH receives hippocampal inputs while other LH subregions do not. Conversely, the tLH is a prominent recipient of nucleus accumbens inputs while the aLH and pLH are not [53]. LH importantly mediates accumbens-elicited food intake [48,54]. Regarding outputs, the aLH innervates thalamic targets, whereas the tLH and pLH preferentially innervate midbrain and hindbrain targets [55]. The tLH to VTA connection in particular is thought to contribute to feeding and self-stimulation [44]. Specific tLH populations project to VTA relevant to feeding, such as LH neurotensin neurons [56]. Other LH galanin neurons project to other targets to influence food intake [57]. These points may be relevant to why the tLH appears to be an especially important subregion for LH feeding and reward-related functions.

In future research, it would be of interest to compare effects of optogenetic excitation to those of optogenetic inhibition of the same LH subregions [58], to explore the relative roles of specific neurochemical/genomic subtypes of neurons within each LH subregions (e.g., neurotensin neurons, [56]), or to explore the role of particular input or output connectivity using a Cre recombinase-expressing canine adenovirus and a Cre-dependent AAV construct [59]. Such approaches would be useful in further delineating the roles of specific neural systems in LH functions.

## Conclusions

In summary, our results help identify differences among LH subregions for supporting optogenetic self-stimulation and evoking laser-bound increases in eating. Specifically, our results anatomically identify the tuberal compartment of LH as potentially most robust for both forms of incentive motivation. Our results also suggest that the behavioral effects of optogenetic LH stimulation appear malleable, and can be changed over time within the same individual by extended laser stimulation, suggesting a flexible motivational process that can alter the shape of stimulation-bound behaviors elicited by optogenetic excitation at a given site.

## Supporting information

**S1 Fig. Sex and virus group difference scores for select behaviors.** Behavioral difference scores (laser period score minus interim period score) for male and female ChR2 and eYFP rats engaging in A. cage crossing during low laser power output or B. grooming during medium laser power output.
(TIF)

## Acknowledgments

We thank our lab managers and technicians who assisted with microscope and animal maintenance over the course of this project, which include Cody Schember, Joshua Goldman, and Nina Mostovoi. Undergraduate students who provided essential assistance in this project by running animals, behavioral video scoring, and capturing microscope images of Fos plumes include Erin Kokoska, Nikit Kapila, Tyler Allerton, and Michelle Nguyen. All procedures used in this laboratory were approved by the Institutional Animal Use and Care Committee of the University of Michigan.

## Author Contributions

**Conceptualization:** Kevin R. Urstadt.

**Formal analysis:** Kevin R. Urstadt.

**Funding acquisition:** Kent C. Berridge.

**Investigation:** Kevin R. Urstadt.

**Methodology:** Kevin R. Urstadt.

**Project administration:** Kent C. Berridge.

**Resources:** Kent C. Berridge.

**Supervision:** Kent C. Berridge.

**Visualization:** Kevin R. Urstadt.

**Writing – original draft:** Kevin R. Urstadt.

**Writing – review & editing:** Kent C. Berridge.

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
