## [Decision Letter · Decision Letter 0]

13 Nov 2019

PONE-D-19-28055

Optogenetic mapping of feeding and self-stimulation within the lateral hypothalamus of the rat

PLOS ONE

Dear Dr. Urstadt,

Thank you for submitting your manuscript to PLOS ONE. After careful consideration, we feel that it has merit but does not fully meet PLOS ONE’s publication criteria as it currently stands. Therefore, we invite you to submit a revised version of the manuscript that addresses the points raised during the review process.

In your submission please address the minor suggestions made by the two reviewers, as listed below. These changes will help clarify some methodological questions and help the reader better understand the nature of the experiments as well as your results. Please be sure to address all reviewer comments.

We would appreciate receiving your revised manuscript by Dec 28 2019 11:59PM. To enhance the reproducibility of your results, we recommend that if applicable you deposit your laboratory protocols in protocols.io, where a protocol can be assigned its own identifier (DOI) such that it can be cited independently in the future. For instructions see: http://journals.plos.org/plosone/s/submission-guidelines#loc-laboratory-protocols

We look forward to receiving your revised manuscript.

Kind regards,

Juan M Dominguez, PhD

Academic Editor

PLOS ONE

Journal Requirements:

1. We note that you have stated that you will provide repository information for your data at acceptance. Should your manuscript be accepted for publication, we will hold it until you provide the relevant accession numbers or DOIs necessary to access your data. If you wish to make changes to your Data Availability statement, please describe these changes in your cover letter and we will update your Data Availability statement to reflect the information you provide.

Reviewers' comments:

Reviewer's Responses to Questions

**Comments to the Author**

1. Is the manuscript technically sound, and do the data support the conclusions?

Reviewer #1: Yes

Reviewer #2: Yes

2. Has the statistical analysis been performed appropriately and rigorously? 

Reviewer #1: Yes

Reviewer #2: Yes

3. Have the authors made all data underlying the findings in their manuscript fully available?

Reviewer #1: Yes

Reviewer #2: Yes

4. Is the manuscript presented in an intelligible fashion and written in standard English?

Reviewer #1: Yes

Reviewer #2: Yes

5. Review Comments to the Author

Reviewer #1: This manuscript by Urstadt and Berridge uses optogenetics to localize the orexigenic effect of the lateral hypothalamus. They provided overwhelming anatomical/behavioral data that the tuberal LH subregion mediates this effect, at least when optogenetics is used as the stimulus. Overall, this is a very logical and well-designed set of experiments. My major concern is that there is an overwhelming amount of data/particularly within each figure, which at times makes the manuscript difficult to follow. I wonder if some of the figures could be moved to a supplemental figure section. And/Or the authors need to do a better job of referring to the individual parts of the figures when describing the supporting statistics in the results section. Some minor issues for clarification:

Methods- Was a total volume of 0.75 µl of virus infused or 0.75 µl per side infused?

In either the methods or results section, it would help if the authors gave rational for doing both both the Laser place preference/avoidance test and the active laser self-stimulation via spout test.

Methods- Page 21 lines 512-528….I don’t think the authors are statistically justified in taking the two outliers and comparing their results with the rest of the group.

Figure 7C- Is this figure missing characters denoting significance? It is discussed in the text.

Reviewer #2: The authors investigate impacts of optogenetic stimulation on subregions of the lateral hypothalamus for food intake, place preference, and self-stimulation. They found optogenetic stimulation parameters sufficient to induce IEG (Fos and Jun) production surrounding the optic fiber tip. Optogenetic stimulation led to subregion specific increases in food intake, place preference, and self-stimulation with the largest effects seen in the tubercle LH. Additionally, in some rats, food intake and self-stimulation was altered after prolonged optogenetic stimulation.

The manuscript would benefit from a discussion comparing the impacts of optogenetic induced feeding behavior and IEG expression after natural feeding. Specifically, do regions with robust optogenetic induced feeding correlate with high levels of IEG expression after natural feeding?

One focus of the manuscript was on changes in behavior following prolonged stimulation exposure. While there was a short discussion of why this stimulation-induced change in behavior might occur, I would encourage the authors to discuss whether this type of strong, prolonged neuronal activation naturally occurs. If so, in what contexts?

For results with sex differences, consider separating sex in figures in order to visualize sex differences. Additionally, consider adding a discussion as to why stimulation of particular regions results in sex differences.

A figure with a timeline of manipulations/behavior would be beneficial for ease of following along.

It has been my experience that prolonged light exposure can cause permanent neuronal damage. Did the authors find evidence for neuronal damage which might impact subsequent behavior?

Did the prolonged stimulation cause seizures?

It was stated that the session ended if a seizure occurred. Did the authors investigate whether the seizure itself caused an animal to subsequently change its self-stimulation or feeding behavior?

Are the effects seen in figure 11A driven by one data point?

6. PLOS authors have the option to publish the peer review history of their article (what does this mean?). If published, this will include your full peer review and any attached files.

Reviewer #1: No

Reviewer #2: Yes: Anne F. Pierce

---

## [Author Response · Author response to Decision Letter 0]

22 Dec 2019

Dear Dr. Dominguez,

We thank you and the Reviewers for the very helpful criticisms and suggestions for our manuscript. We are pleased that this work may be of interest to readers of PLoS One, once revisions are addressed. As such, we have now revised the manuscript in line with the reviewer-suggested changes to enhance its quality.

Reviewer #1 Comments: 

Reviewer: This manuscript by Urstadt and Berridge uses optogenetics to localize the orexigenic effect of the lateral hypothalamus. They provided overwhelming anatomical/behavioral data that the tuberal LH subregion mediates this effect, at least when optogenetics is used as the stimulus. Overall, this is a very logical and well-designed set of experiments. My major concern is that there is an overwhelming amount of data/particularly within each figure, which at times makes the manuscript difficult to follow. I wonder if some of the figures could be moved to a supplemental figure section. And/Or the authors need to do a better job of referring to the individual parts of the figures when describing the supporting statistics in the results section. 

Response: We are grateful that Reviewer 1 thought these were logical and well-designed experiments. We appreciate that the figures are dense, especially because they unpack data into anatomical subdivisions of the LH into 3 separate planes (i.e., A-P; M-L; D-V). However, we believe that this provides the reader with the most complete picture of the real data pattern, showing localizations of function and revealing functional differences between anatomical subdivisions. To help make particular data in figures more interpretable and easy to find, we have now inserted several more specific descriptions of relevant data locations within figures in Results. We have also added stereotaxic group labels to bar graphs in the mapping figures (Fig 6, 8 & 11) as well as background gray “bounding” boxes to more clearly group stereotaxic groups in dense bar graph figures (Fig 7, 9 & 10).

Reviewer: Some minor issues for clarification: Methods- Was a total volume of 0.75 µl of virus infused or 0.75 µl per side infused?

Response: The volume of virus infused was 0.75 uL per hemisphere (thus 1.5 uL per rat). We have changed the text to clarify this at line 129-130 on page 6, and we thank the Reviewer for drawing this to our attention.

Reviewer: In either the methods or results section, it would help if the authors gave rational for doing both both the Laser place preference/avoidance test and the active laser self-stimulation via spout test.

Response: We have now included the rationale in the Methods in lines 192-210, starting on page 8. The text reads:

“We utilized this place preference/avoidance test in addition to the spout self-stimulation test as each evaluates the motivational valence of laser stimulation in somewhat different ways. The place preference/avoidance test tends to deliver more prolonged or repeated laser stimulation, because a new 1 s stimulation is triggered by each movement within the laser-administering corner. This provides a rat with an easy way to bask in prolonged laser stimulation if it is rewarding. The place test also allows the detection of avoidance of aversive laser properties, again especially if these emerge with more dense exposure to laser stimulation. Finally, the place test is sensitive to Pavlovian conditioning of preference or avoidance to the location as conditioned stimulus. The spout self-stimulation test, on the other hand, gives a rat more precise instrumental control of laser stimulation. It requires a new active instrumental response (spout touch) to trigger a laser stimulation. It also provides the rat with greater control over the amount and timing of stimulation. The spout test typically delivers less dense laser stimulation, allowing best detection of reward/incentive effects, especially if laser becomes more negatively aversive with greater density. And the spout is directly sensitive to detecting traditional instrumental response reinforcement, that is, strengthening the probability of an immediately preceding and action. For these reasons, we believe that both tests of self-stimulation have value. And when both give the same answer, they establish the generality of positive incentive effects of laser by showing that self-stimulation is not limited to the parameters of one particular paradigm.”

Reviewer: Methods- Page 21 lines 512-528….I don’t think the authors are statistically justified in taking the two outliers and comparing their results with the rest of the group.

Response: Our reasoning is that these “outliers” may actually be the most stable stimulation-bound eaters, serving as individualized “case-studies”. That is, there may be stable individual differences of LH optogenetic stimulation effects even within a single subregion. For example, it was well established in the 1960s to 1980s that only some individuals were reliable stimulation-bound eaters in LH electrical stimulation studies (e.g., by Neal Miller; James Olds; Elliot Valenstein; etc.). We also elaborate on our rationale for focusing on these outliers in the Discussion on page 36, lines 887-892 (copied here):

“In our study, it also seems to be the case that certain individuals are reliable laser-bound eaters, while others are not, even for the same LH subregion. That is, while particular LH subregions have a much higher probability of supporting laser-induced eating than other subregions, the probability is not 100% across all individuals. We are trying to recognize such individual differences, in hope that future research may help reveal what individual features beyond LH subregion contribute to controlling stimulation-bound eating.” 

Reviewer: Figure 7C- Is this figure missing characters denoting significance? It is discussed in the text.

Response: Note that the relevant figure is now 8C. The Reviewer is correct in that no significance symbol is noted on the figure. There was not a significant difference in the change of laser corner beam breaks, from pre-experience to post-experience, between different stereotaxic groups. There is however a significant difference the laser corner beam breaks for the tLH group before versus after laser experience. This is now marked with significance markers, for example the “#” symbol as in Figure 9 (relevant to line 587 in manuscript). 

Reviewer #2 Comments: 

Reviewer: The authors investigate impacts of optogenetic stimulation on subregions of the lateral hypothalamus for food intake, place preference, and self-stimulation. They found optogenetic stimulation parameters sufficient to induce IEG (Fos and Jun) production surrounding the optic fiber tip. Optogenetic stimulation led to subregion specific increases in food intake, place preference, and self-stimulation with the largest effects seen in the tubercle LH. Additionally, in some rats, food intake and self-stimulation was altered after prolonged optogenetic stimulation.

The manuscript would benefit from a discussion comparing the impacts of optogenetic induced feeding behavior and IEG expression after natural feeding. Specifically, do regions with robust optogenetic induced feeding correlate with high levels of IEG expression after natural feeding?

Response: We agree that correlation of optogenetically-effective subregions with natural eating or reward activations is an interesting issue to discuss. Accordingly, we have now added some discussion of this natural activation issue in a new paragraph starting on page 35, line 872-886, and continuing on line 910-916 on page 37.

Reviewer: One focus of the manuscript was on changes in behavior following prolonged stimulation exposure. While there was a short discussion of why this stimulation-induced change in behavior might occur, I would encourage the authors to discuss whether this type of strong, prolonged neuronal activation naturally occurs. If so, in what contexts?

Response: Regarding changes in optogenetically-induced behaviors from the same site following prolonged laser stimulation, it is not our assertion that such prolonged LH activation necessarily occurs in natural situations. It might or might not, and it is difficult to find relevant data in the literature. However, our main purpose is to elucidate the nature of stimulation-bound behaviors, and to provide a cautionary note for brain stimulation studies. That is, behavioral effects induced by optogenetic stimulation or DREADD stimulation, etc., are often reported in neuroscience literature with an apparent assumption that observed effects are intrinsic to the particular stimulated neural system, and likely to be stable and permanent. There is typically not recognition of any possibility that behavioral effects of neural stimulation can change even for the same site, same intensity, etc. We offer our observation that LH optogenetic effects on behavior can change with prolonged stimulation with two intentions: 1) to demonstrate the potential functional flexibility of LH circuitry, and 2) as a cautionary example against assuming that specific functions observed during stimulation are necessarily permanent or intrinsic. We have now inserted statements briefly discussing these ideas in lines 910-916. 

Reviewer: For results with sex differences, consider separating sex in figures in order to visualize sex differences. Additionally, consider adding a discussion as to why stimulation of particular regions results in sex differences.

Response: We agree, and so have now added sex average value indicators to data in Fig 9B in the AP location column, Fig 9C in the AP location column, and in Figure 15B. Also, we have added a separate Fig S1 comparing behavioral difference scores between sexes and virus groups for select behaviors. Lastly, we have added a summary of these results to the Discussion in lines 865-871 starting on page 35.

Reviewer: A figure with a timeline of manipulations/behavior would be beneficial for ease of following along.

Response: We thank the reviewer for this suggestion, and have now added a timeline figure in Methods, referenced at line 111.

Reviewer: It has been my experience that prolonged light exposure can cause permanent neuronal damage. Did the authors find evidence for neuronal damage which might impact subsequent behavior?

Response: We did not observe evidence of light damage, although we did not use a TUNEL assay to directly quantify it, nor did we include no-light surgical controls that received virus and implants but were never illuminated. Those might be needed in order to detect light-induced damage (as separate from gliosis surrounding virus microinjection site or optic fiber) or to conclude with confidence that no light-induced damage occurred. In tissue sections containing the fiber tract and virus expression below it, tissue appeared normal aside from gliosis immediately surrounding the perimeter of the fiber tract, as would be the case with any brain implant. However we stress that we use relatively low power laser illumination (2-3 mW/mm2), compared to some other optogenetic studies (10-40 mW/mm2), which might account for why we do not see detectable damage. We find the low power to be sufficiently behaviorally effective, and wish to avoid intensities that are higher than needed. Neuronal damage may be less likely at the lower illumination intensities we used.

We are also aware that blue light can increase markers of inflammation; Tyssowski and Gray (2019; https://doi.org/10.1523/ENEURO.0085-19.2019) discuss this effect in multiple contexts, but in their in vitro testing they only see elevation of certain genes related to oxidation after 1+ hours of constant blue light administration, albeit at 2-4 mW/mm2. They do not mention increases in apoptosis resulting from light application. In our experiments, even over the span of the prolonged laser experience sessions, the total amount of stimulation would amount to 12 min of pulsed low-intensity light distributed across a 2 hr session in one day in an in vivo situation. Thus, we do not find significant apoptotic effects on neurons in this project, although we agree the Reviewer’s concern about damage is valid, and we acknowledge that our oxidative consequences remain unknown. 

Reviewer: Did the prolonged stimulation cause seizures?

It was stated that the session ended if a seizure occurred. Did the authors investigate whether the seizure itself caused an animal to subsequently change its self-stimulation or feeding behavior?

Response: No, we did not observe any seizures or pre-seizure activity in LH or LPO rats, even during the behavioral assays when the highest laser power output was applied. Our lab has observed seizures from optogenetic stimulation of other brain regions (e.g., amygdala or insula cortex), so we include watchfulness for behavioral evidence of seizures as a standard procedure. But we did not see any such activity, and we have now inserted mention of that in text; see line 221-225 on pages 9-10.

Reviewer: Are the effects seen in figure 11A driven by one data point?

Response: The one data point is a major contributor, as without it the correlation was no longer significant. We now mention this on page 29, lines 700-702.

Again, we thank the Reviewers for their helpful suggestions. We believe these changes in response improve the revised manuscript, and we hope you will agree.

Sincerely,

Kevin Urstadt and Kent Berridge

---

## [Editor Report · Decision Letter 1]

30 Dec 2019

Optogenetic mapping of feeding and self-stimulation within the lateral hypothalamus of the rat

PONE-D-19-28055R1

Dear Dr. Urstadt,

We are pleased to inform you that your manuscript has been judged scientifically suitable for publication and will be formally accepted for publication once it complies with all outstanding technical requirements.

With kind regards,

Juan M Dominguez, PhD

Academic Editor

PLOS ONE

---

## [Editor Report · Acceptance letter]

7 Jan 2020

PONE-D-19-28055R1 

Optogenetic mapping of feeding and self-stimulation within the lateral hypothalamus of the rat 

Dear Dr. Urstadt:

I am pleased to inform you that your manuscript has been deemed suitable for publication in PLOS ONE. Congratulations! Your manuscript is now with our production department. 

With kind regards,

on behalf of

Dr Juan M Dominguez 

Academic Editor

PLOS ONE